# Sequential Latent Variable Models for Few-Shot High-Dimensional Time-Series Forecasting

**Xiajun Jiang,**[*] **Ryan Missel,**[*] **Zhiyuan Li & Linwei Wang**
Golisano College of Computing and Information Sciences
Rochester Institute of Technology
Rochester, NY 14623, USA
`{xj7056,rxm7244,zl7904,Linwei.Wang}@rit.edu`

## Abstract

Modern applications increasingly require learning and forecasting latent dynamics from high-dimensional time-series. Compared to univariate time-series forecasting, this adds a new challenge of reasoning about the latent dynamics of an *unobserved* abstract state. Sequential latent variable models (SLVMs) present an attractive solution, although existing works either struggle with long-term forecasting or have difficulty learning across diverse dynamics. In this paper, we first present a conceptual framework of SLVMs to unify existing works, contrast their fundamental limitations, and identify an intuitive solution to *long-term forecasting for diverse dynamics* via meta-learning. We then present a few-shot forecasting framework for high-dimensional time-series: instead of learning a single dynamic function, we leverage data of diverse dynamics and learn to adapt latent dynamic functions to few-shot support series. This is realized via Bayesian meta-learning underpinned by: 1) a latent dynamic function conditioned on knowledge derived from few-shot support series, and 2) a meta-model that learns to extract such dynamic-specific knowledge via feed-forward embedding of support set. We compared the presented framework with a comprehensive set of baseline models 1) trained globally on the large meta-training set with diverse dynamics, 2) trained individually on single dynamics with and without fine-tuning to $k$-shot support series, and 3) extended to few-shot meta-formulations. We demonstrated that the presented framework is agnostic to the latent dynamic function of choice and, at meta-test time, is able to forecast for new dynamics given variable-shot of support series.[1]

## 1 Introduction

In many applications, an ultimate goal is to forecast the future states or trajectories of a dynamic system from its high-dimensional observations such as series of images. Compared to the relatively well-studied *univariate* time-series forecasting (Makridakis et al., 2018; Oreshkina et al., 2020; Salinas et al., 2020), *high-dimensional* time-series forecasting raises new challenges: it requires the extraction of the dynamics of an abstract latent state that is not directly observed (Botev et al., 2021).

*Sequential latent variable models* (SLVMs) provide an attractive solution that, unlike autoregressive models, abstracts a latent dynamic function $\mathbf{z}_i = f(\mathbf{z}_{<i}; \boldsymbol{\theta}_z)$ with state $\mathbf{z}_i$ and parameter $\boldsymbol{\theta}_z$, along with $\mathbf{z}_i$'s emission to observations $\mathbf{x}_i = g(\mathbf{z}_i)$ (Chung et al., 2015). This pair of learned models can support long-term forecasting given only initial frames of observations, as well as controlled generation of new dynamics. Critical bottlenecks however remain in reaching these goals.

The earlier formulation of SLVMs relies on a natural extension of the static LVMs: as illustrated in Fig. 1A, the latent state $\mathbf{z}_i$ is modeled as the latent variable for the generation of $\mathbf{x}_i$, and a sequential encoder is used to facilitate the inference of $\mathbf{z}_i$ from current and past observations $\mathbf{x}_{\leq i}$ (Chung et al., 2015; Krishnan et al., 2017). Recent works have argued to instead model and infer the parameter

---

[*]Both authors contributed equally to this work
[1]Source code available at `https://github.com/john-x-jiang/meta_ssm`.

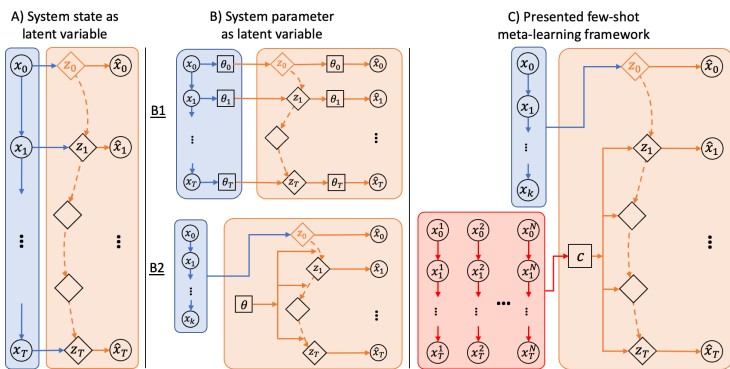

Figure 1: Sequential latent-variable models for forecasting high-dimensional sequences.

of the latent dynamic function, often modeled as time-varying linear coefficients $\boldsymbol{\theta}_{z,i}$ (Karl et al., 2017; Fraccaro et al., 2017; Rangapuram et al., 2018; Klushyn et al., 2021). This results in an LVM formulation as illustrated in Fig. 1B1, where the latent variable $\boldsymbol{\theta}_{z,i}$ is inferred at each $i$ from observations $\mathbf{x}_{\leq i}$. While strong at time-series reconstructions and classifications, a fundamental limitation makes these SLVMs less suited for long-term forecasting: the latent dynamic function has a limited ability to forecast without near-term observations to support the inference of $\mathbf{z}_i$ or $\boldsymbol{\theta}_{z,i}$.

This limitation in the mainstream SLVMs raises a natural question: are we able to relax the assumption of linear dynamic function and directly infer its $\boldsymbol{\theta}_z$? Works adopting this idea have emerged: as illustrated in Fig. 1B2, by modeling a single $\boldsymbol{\theta}_z$ – either deterministic (Rubanova et al., 2019; Botev et al., 2021) or stochastic (Yildiz et al., 2020) – $f(\mathbf{z}_{<i}; \boldsymbol{\theta}_z)$ can be asked to predict a time sequence using only an inferred initial state. This formulation has shown strong long-term forecasting, although with its own fundamental limitation: it learns a *single dynamic function* global to all training sequences. This would not only require all training data to share identical latent dynamics, but also has difficulty to forecast test sequences with dynamics different from or unknown to the training.

In this paper, we answer this important open question of *long-term forecasting for diverse dynamics*. We first present a conceptual framework of SLVMs to unify existing works, and identify an intuitive solution to the underlying critical gap via meta-learning: instead of *learning a single dynamic function*, we can learn to pull knowledge across datasets of different dynamics and *learn to adapt a dynamic function* to few-shot high-dimensional time-series. We then present a Bayesian meta-learning framework as illustrated in Fig. 1C: instead of being a single fixed function as in Fig. 1B2, we let the latent dynamic function be conditioned on knowledge derived from few-shot support time-series via a feed-forward set-embedding meta-model; given $k$-shot time-series of a specific dynamics, the model is asked to forecast for query time-series using only the initial frames, meta-learned across dynamics. We develop this framework to be agnostic to the latent dynamic functions of choice, and with the flexibility to forecast with a variable size of $k$.

We evaluated the presented framework in benchmark image sequences with mixed physics including bouncing balls (Fraccaro et al., 2017), pendulum (Botev et al., 2021), and mass-spring (Botev et al., 2021). We further applied it to forecasting complex physics of turbulence flow (Wang et al., 2021) and electrical dynamics over 3D geometrical meshes of the heart. We compared the presented work with SLVMs representative of each of the formulations in Fig. 1A-B, along with a recent autoregressive model designed to forecast for diverse dynamics (Donà et al., 2020). Each baseline model was trained on 1) the large meta-training set with diverse dynamics, and 2) each dynamics individually, both with and without fine-tuning to $k$-shot support data. Representative SLVMs were further tested in their feed-forward or optimization-based meta-extensions. Results demonstrated clear margins of improvements by the presented work in forecasting diverse dynamics, with added ability to recognize clusters of distinct dynamics and allow controlled time-series generation given only initial conditions.

## 2 RELATED WORKS & BACKGROUND

**Sequential LVMs:** Among the first SLVMs is the variational recurrent neural networks (VRNN) (Chung et al., 2015), followed by a series of deep state-space models (SSMs) (Krishnan et al., 2017;

Maddison et al., 2017; Li et al., 2019) focused on modeling the dependence of the posterior and transitional density of the latent state $\mathbf{z}_k$ on past latent states $\mathbf{z}_{<k}$ and observations $\mathbf{x}_{<k}$ (Fig. 1A) – resembling the deep extensions of the classic Kalman filter (Krishnan et al., 2017) and particle filter (Maddison et al., 2017). An alternative line of deep SSMs aims to infer the parameters of the latent dynamic function instead (Karl et al., 2017; Fraccaro et al., 2017; Rangapuram et al., 2018; Klushyn et al., 2021). Existing approaches along this line assumed linear latent dynamics, where the linear transition matrix at each time frame $k$ is modeled as a linear combination of a set of global matrices. The linear coefficients are modeled to be time-varying and inferred from observations $\mathbf{x}_{\leq k}$ as illustrated in Fig. 1B1. In both formulations, the latent dynamic function's reliance on inferred time-varying variables reduces its ability to forecast without near-term observations.

In parallel, a set of models (Fig. 1B2) have been presented that aims to learn a latent dynamic function that forecasts a sequence using only an inferred initial state, in stochastic (Rubanova et al., 2019; Yildiz et al., 2020) or deterministic forms (Botev et al., 2021). The resulting latent dynamic function is strong at forecasting, albeit only a single function is learned at a time. We build on and advance this formulation of learning to learn a dynamic-specific function from few-shot observations.

Modeling switching dynamics in SLVMs, often based on the formulation in Fig. 1A, shares the presented idea of using context variables to control the latent dynamics (Becker-Ehmck et al., 2019; Linderman et al., 2017). They however are concerned with the switching of dynamics within a time-series, whereas we are interested in learning to learn dynamics from $k$-shot support series.

Sequential neural processes (SNPs), based on SLVM formulation in Fig. 1A (Singh et al., 2019; Qin et al., 2019), are underlined by Bayesian meta-learning similar to the presented work. They are originally designed for supervised learning of a regression function over time instead of forecasting. In this work, we will extend SNP to realize a meta-version of the SLVM formulation in Fig. 1A, as a counterpart to be compared with the presented meta-SLVM in Fig. 1C.

**Autoregressive dynamics:** Autoregressive models are also popular for modeling and forecasting dynamics, especially for approximating physics-based simulations (Wang et al., 2020; Pfaff et al., 2020). Some recent works have focused on generalizing across dynamics by, for instance, disentangling spatial and temporal modeling (Donà et al., 2020) or learning dynamic-specific functions in addition to a global dynamic function (Yin et al., 2021). A recent autoregressive model considered "meta-learning" dynamics by using task-embedding to condition the forecaster (Wang et al., 2021), although this task encoder is trained separately from the forecasting model via weak supervision and it infers the task from the observed frames of a forecasting series. Moreover, autoregressive models cannot support controlled generation of time-series as we will demonstrate in Section 5.

**General few-shot learning:** Few-shot learning has seen substantial progress with static data, including weight initialization (Finn et al., 2017; Yoon et al., 2018), model optimizers (Ravi & Larochelle, 2016), and feed-forward models to condition (Garnelo et al., 2018) or parameterize the primary networks (Bertinetto et al., 2016; Sung et al., 2018). Among these, feed-forward meta-models replace test-time optimization with simple feed-forward passes using support data. It also has an interesting high-level relation to Exemplar VAE (Norouzi et al., 2020) where the few-shot support samples can be viewed as the exemplar. It thus constitutes the basis of the presented few-shot forecasting methods.

**Few-shot time-series forecasting:** Meta-learning is well studied in univariate time-series forecasting (Montero-Manso et al., 2020) including recent deep-learning advances (Oreshkin et al., 2021). Few-shot forecasting for high-dimensional time-series, however, has not been attempted to our knowledge.

## 3 UNIFYING CONCEPTUAL FRAMEWORK FOR LEARNING LATENT DYNAMICS

We first describe an LVM framework that unifies existing works under two choices of probabilistic graphical models (PGMs). It includes a dynamic function of latent $\mathbf{z}_k$ and its emission to data $\mathbf{x}_k$: $\mathbf{z}_k = f(\mathbf{z}_{<k}; \boldsymbol{\theta}_z), \mathbf{x}_k = g(\mathbf{z}_k)$, where $\boldsymbol{\theta}_z$ represents the parameter of the latent dynamic function.

**System states as latent variables:** One natural choice of the latent variable is the latent state $\mathbf{z}_k$ underlying the observations $\mathbf{x}_k$. This gives rise to the PGM as illustrated in Fig. 1A, where the marginal likelihood of an observed sequence $\mathbf{x}_{0:T}$ can be expressed as:

$$p(\mathbf{x}_{0:T}) = \int_{\mathbf{z}_{0:T}} p(\mathbf{x}_0|\mathbf{z}_0)p(\mathbf{z}_0) \prod_{i=1}^{T} p(\mathbf{x}_i|\mathbf{z}_i)p(\mathbf{z}_i|\mathbf{z}_{<i}, \mathbf{x}_{<i})d\mathbf{z}_{0:T}, \tag{1}$$

where $p(\mathbf{x}_i|\mathbf{z}_i)$ describes emission and $p(\mathbf{z}_i|\mathbf{z}_{<i}, \mathbf{x}_{<i})$ describes latent dynamics. To facilitate inference, a variational approximation of the posterior density $q(\mathbf{z}_{0:T}|\mathbf{x}_{0:T})$ is often modeled as $q(\mathbf{z}_{0:T}|\mathbf{x}_{0:T}) = \prod_{i=1}^{T} q(\mathbf{z}_i|\mathbf{z}_{<i}, \mathbf{x}_{\leq i})$. The evidence lower bound (ELBO) of Equation (1) is:

$$\log p(\mathbf{x}_{0:T}) \geq \sum_{i=0}^{T} \mathbb{E}_{q(\mathbf{z}_i|\mathbf{z}_{<i}, \mathbf{x}_{\leq i})}\left[\log p(\mathbf{x}_i|\mathbf{z}_i)\right] - \mathrm{KL}(q(\mathbf{z}_i|\mathbf{z}_{<i}, \mathbf{x}_{\leq i})||p(\mathbf{z}_i|\mathbf{z}_{<i}, \mathbf{x}_{<i})). \quad (2)$$

Existing works adopting this PGM (Chung et al., 2015; Krishnan et al., 2017; Li et al., 2019) differ primarily in how $p(\mathbf{z}_i|\mathbf{z}_{<i}, \mathbf{x}_{<i})$ and $q(\mathbf{z}_i|\mathbf{z}_{<i}, \mathbf{x}_{\leq i})$ are modeled. The first term above encourages reconstruction using the inferred $q(\mathbf{z}_i|\mathbf{z}_{<i}, \mathbf{x}_{\leq i})$ at each time frame $i$; this weakens the latent dynamic function underlying $p(\mathbf{z}_i|\mathbf{z}_{<i}, \mathbf{x}_{<i})$ that is constrained only by the KL-divergence term. This leads to limited ability to forecast without near-term $\mathbf{x}_{\leq i}$ to support the inference of $q(\mathbf{z}_i|\mathbf{z}_{<i}, \mathbf{x}_{\leq i})$.

**System parameters as latent variables:** An alternative choice of the latent variable is the parameters themselves of the LVM equation, especially $\boldsymbol{\theta}_z$ of the latent dynamic function. This gives rise to the PGM in Fig. 1B, where the marginal likelihood of $\mathbf{x}_{0:T}$ can now be expressed as:

$$p(\mathbf{x}_{0:T}) = \int_{\mathbf{z}_0} p(\mathbf{x}_0|\mathbf{z}_0)p(\mathbf{z}_0)d\mathbf{z}_0 \int_{\boldsymbol{\theta}_z} \prod_{i=1}^{T} p(\mathbf{x}_i|\mathbf{z}_i)|_{\mathbf{z}_i = f(\mathbf{z}_{<i};\boldsymbol{\theta}_z)} p(\boldsymbol{\theta}_z)d\boldsymbol{\theta}_z, \quad (3)$$

where the observations are explained by an initial latent state $\mathbf{z}_0$ and parameter $\boldsymbol{\theta}_z$ of the latent dynamic function. With a variational approximation of the posterior density $q(\boldsymbol{\theta}_z, \mathbf{z}_0)$ and an assumption of their prior densities $p(\mathbf{z}_0)$ and $p(\boldsymbol{\theta}_z)$, the ELBO of Equation (3) becomes:

$$\log p(\mathbf{x}_{0:T}) \geq \mathbb{E}_{q(\boldsymbol{\theta}_z, \mathbf{z}_0)}\left[\log p(\mathbf{x}_{0:T}|\mathbf{z}_0, \boldsymbol{\theta}_z)\right] - \mathrm{KL}(q(\mathbf{z}_0)||p(\mathbf{z}_0)) - \mathrm{KL}(q(\boldsymbol{\theta}_z)||p(\boldsymbol{\theta}_z)), \quad (4)$$

This covers different lines of existing works depending on how $q(\boldsymbol{\theta}_z)$ and $p(\boldsymbol{\theta}_z)$ are modeled. In a series of works (Karl et al., 2017; Fraccaro et al., 2017; Rangapuram et al., 2018; Klushyn et al., 2021), $\boldsymbol{\theta}_z$ is modeled as time-varying system parameters $\boldsymbol{\theta}_{z,0:T}$. This involves intricate temporal modeling of $q(\boldsymbol{\theta}_{z,i}|\mathbf{x}_{\leq i})$ and $p(\boldsymbol{\theta}_{z,i}|\mathbf{z}_{\leq i})$ over time as illustrated in Fig. 1B1. Because the latent dynamic function relies on time-varying $\boldsymbol{\theta}_{z,0:T}$, its forecasting again relies on near-term observations to support the inference of $\boldsymbol{\theta}_{z,i}$. Alternatively, $q(\boldsymbol{\theta}_z)$ can be simply assumed to be global across observations and the dynamic function becomes a Bayesian neural network as presented by (Yildiz et al., 2020). As a more special case, $\boldsymbol{\theta}_z$ can be deterministic which leads to the latent ODE model presented by (Rubanova et al., 2019). If we further assume $\mathbf{z}_0$ to be deterministic, we arrive at the set of deterministic encoding-decoding network with latent dynamic functions examined by (Botev et al., 2021). This set of formulations, as summarized in Fig. 1B2, shares the advantage of strong long-term forecasting, albeit a fundamental limitation in learning a single dynamic function at a time.

In Section 5, we will include representative models from each PGM to provide empirical evidence for the identified limitations. With this basis, we derive an intuitive solution to the identified critical gaps by extending the PGM in Fig. 1B2 to the presented PGM in Fig. 1C: instead of learning a single dynamic function, we will learn to adapt a latent dynamic function to few-shot support time-series.

## 4 FEW-SHOT FORECASTING VIA BAYESIAN META-LEARNING

Consider a dataset $\mathcal{D}$ of high-dimensional time-series with $M$ similar but distinct underlying dynamics: $\mathcal{D} = \{\mathcal{D}_j\}_{j=1}^{M}$. For each $\mathcal{D}_j$, we consider disjoint few-shot *support* series $\mathcal{D}_j^s = \{\mathbf{x}_{0:T}^{s,1}, \mathbf{x}_{0:T}^{s,2}, ..., \mathbf{x}_{0:T}^{s,k}\}$ and *query* series $\mathcal{D}_j^q = \{\mathbf{x}_{0:T}^{q,1}, \mathbf{x}_{0:T}^{q,2}, ..., \mathbf{x}_{0:T}^{q,l}\}$ where $k \ll l$. Instead of maximizing the marginal likelihood of $\mathbf{x}_{0:T}$ for all $\mathbf{x}_{0:T} \in \mathcal{D}$ as in Equation (3), we formulate a meta-objective to learn to maximize the marginal likelihood of $\mathbf{x}_{0:T}^q$ for all query series $\mathbf{x}_{0:T}^q \in \mathcal{D}_j^q$ when conditioned on support series $\mathcal{D}_j^s$, for all dynamics $j \in \{1, 2, ..., M\}$:

$$p(\mathbf{x}_{0:T}^q|\mathcal{D}_j^s) = \int_{\mathbf{c}} p(\mathbf{x}_{0:T}^q|\mathbf{c})p(\mathbf{c}|\mathcal{D}_j^s)d\mathbf{c}, \quad \mathbf{x}_{0:T}^q \in \mathcal{D}_j^q \quad (5)$$

where $p(\mathbf{x}_{0:T}^q|\mathbf{c})$, though similar to Equation (3), is now conditioned on (thus adapted to) knowledge derived from support series of a specific dynamics. $p(\mathbf{c}|\mathcal{D}_j^s)$ is the meta-model describing how to extract such dynamic-specific knowledge from few-shot support set $\mathcal{D}_j^s$.

**Set-conditioned latent dynamic functions:** We model $p(\mathbf{x}_{0:T}^q|\mathbf{c})$ based on Equation (3) as:

$$p(\mathbf{x}_{0:T}^q|\mathbf{c}) = \int_{\mathbf{z}_0} p_{\theta_x}(\mathbf{x}_0|\mathbf{z}_0)p(\mathbf{z}_0)d\mathbf{z}_0 \prod_{i=1}^{T} p_{\theta_x}(\mathbf{x}_i|\mathbf{z}_i)|_{\mathbf{z}_i = f(\mathbf{z}_{i-1}, \mathbf{c}; \boldsymbol{\theta}_z)}, \quad (6)$$

where the latent dynamic function is parameterized by $\boldsymbol{\theta}_z$ but conditioned on embedding $\mathbf{c}$ from the support set. To focus on $\mathbf{c}$, we assume $\boldsymbol{\theta}_z$ to be deterministic and global as in (Rubanova et al., 2019; Botev et al., 2021). As an example, we can describe $\mathbf{z}_i = \tilde{\mathbf{z}}_{i-1} + \Delta\mathbf{z}_i$ where $\Delta\mathbf{z}_i$ conditions gated recurrent units (GRUs) (Chung et al., 2014) on $\mathbf{c}$, as detailed in Appendix B. This conditioning can be generalized to other functional forms of $f(\cdot)$, which we will demonstrate in experiments.

**Meta-model for amortized variational inference:** We model $p_\zeta(\mathbf{c}|\mathcal{D}_j^s)$ with a meta-model parameterized by $\zeta$ in the form of feed-forward embedding of support set $\mathcal{D}_j^s$. Specifically, each support sequence $\mathbf{x}_{0:T}^s \in \mathcal{D}_j^s$ is first encoded through a neural function $h_\phi(\mathbf{x}_{0:T}^s)$ with blocks of interlaced spatial convolution and temporal compression layers. To extract knowledge shared by the set, the embedding from all sequences in $\mathcal{D}_j^s$ is aggregated by an averaging function: $\frac{1}{k}\sum_{\mathbf{x}_{0:T}^s \in \mathcal{D}_j^s} h_\phi(\mathbf{x}_{0:T}^s)$, where $k$ is the size of the support set. The value of $k$ can be fixed or variable in our framework. This set embedding parameterizes $p_\zeta(\mathbf{c}|\mathcal{D}_j^s) \sim \mathcal{N}(\boldsymbol{\mu}_c, \boldsymbol{\sigma}_c^2)$ via separate linear layers.

To enable inference, we approximate the posterior density $p(\mathbf{c}|\mathcal{D}_j^s, \mathbf{x}_{0:T}^q)$ as $q_\zeta(\mathbf{c}|\mathcal{D}_j^s \cup \mathbf{x}_{0:T}^q)$, sharing the same meta set-embedding model by augmenting $\mathcal{D}_j^s$ with $\mathbf{x}_{0:T}^q$. The ELBO of Equation (5) across all dynamics $\mathcal{D} = \{\mathcal{D}_j\}_{j=1}^M$ can then be derived as:

$$\sum_{j=1}^M \sum_{\mathbf{x}_{0:T}^q \in \mathcal{D}_j^q} \log p(\mathbf{x}_{0:T}^q|\mathcal{D}_j^s) \quad \geq \sum_{j=1}^M \sum_{\mathbf{x}_{0:T}^q \in \mathcal{D}_j^q} \mathbb{E}_{q_\phi(\mathbf{z}_0^q), q_\zeta(\mathbf{c}|\mathcal{D}_j^s \cup \mathbf{x}_{0:T}^q)}[\log p_{\theta_x}(\mathbf{x}_{0:T}^q|\mathbf{z}_0^q, \mathbf{c})] \quad (7)$$

$$-\text{KL}(q_\phi(\mathbf{z}_0^q|\mathbf{x}_{0:l_{z_0}}^q)||p(\mathbf{z}_0)) - \text{KL}\left(q_\zeta(\mathbf{c}|\mathcal{D}_j^s \cup \mathbf{x}_{0:T}^q)||p_\zeta(\mathbf{c}|\mathcal{D}_j^s)\right),$$

where $q_\phi(\mathbf{z}_0^q|\mathbf{x}_{0:l_{z_0}}^q) \sim \mathcal{N}(\boldsymbol{\mu}_{z_0}, \boldsymbol{\sigma}_{z_0}^2)$ is parameterized by an encoder with $l_{z_0} = 2$ in all experiments. $p(\mathbf{z}_0)$ is assumed to be $\mathcal{N}(0, \mathbf{I})$. The likelihood term is estimated with reparameterization trick (Kingma & Welling, 2013), and the KL-divergence terms are calculated analytically.

The optimization of Equation (7) is realized via episodic training where, in each training episode, data in each dynamic set $\mathcal{D}_j$ is divided into disjoint support set $\mathcal{D}_j^s$ and query set $\mathcal{D}_j^q$. For each query series across all dynamics, starting with an initial latent state $\mathbf{z}_0$ (inferred from $l_{z_0}$ frames) and $k$-shot support embedding $\mathbf{c}$, the latent dynamic function is asked to propagate forward to forecast the entire sequence of $\mathbf{z}_{0:T}$ and their corresponding high-dimensional observations $\mathbf{x}_{0:T}$.

## 5 EXPERIMENTS ON BENCHMARK IMAGE SEQUENCES

**Data:** We first considered benchmark images generated with controllable physics, including bouncing ball Fraccaro et al. (2017), Hamiltonian pendulum (Botev et al., 2021), and Hamiltonian mass-spring systems (Botev et al., 2021). Details of data generation are available in Appendix G. To intentionally create data with diverse dynamics, we included 1) a bouncing ball dataset with 16 different directions of gravity, each with 3000 samples simulated using a combination of different initial positions and velocities (*gravity-16*); and 2) a mixed-physics dataset consisting of bouncing balls under 4 gravity directions, and pendulums and mass springs each with four different values of friction coefficients of $0, 0.05, 0.1, 0.15$ (*mixed-physics*). Each physics with a unique parameter includes 3000 samples.

**Models:** We considered baseline models representative of each formulation outlined in Fig. 1. This includes VRNN Chung et al. (2015) and DKF Krishnan et al. (2017) representing Fig. 1A1, DVBF Karl et al. (2017) and KVAE Fraccaro et al. (2017) representing Fig. 1B1, and three models representing Fig. 1B2 with latent dynamic functions as residual GRUs (GRU-res), neural ordinary differential equation (NODE), and residual Recurrent Generative Networks (RGN-res) (Botev et al., 2021). We also considered a recent autoregressive model designed to tackle forecasting diverse dynamics (Donà et al., 2020). All baseline models were 1) trained using the entire meta-training data consisting of mixed dynamics, 2) trained in 1) and further fine-tuned to the meta-test $k$-shot support set ($k = 15$) (except for (Donà et al., 2020) as we were uncertain about a proper approach of fine-tuning due to its specialized architecture), and 3) trained individually for each single dynamics, with and without fine-tuning to the meta-test $k$-shot support set ($k = 15$).

For each of the global latent dynamic models (GRU-res, NODE, and RGN-res), we extended it into our few-shot framework. While few-shot learning with the rest of the SLVMs is not yet reported in literature, we further selected DKF as a representative of the SLVM in Fig. 1A and extended it into a feed-forward meta-formulation via a variant of the SNP (meta-DKF). We also attempted

Table 1: Comparison of the presented meta-models with all baselines trained on the meta-training set for gravity-16 data. The improvement of meta-GRU-res (best-performing) over its closest baseline is statistically significant in all metrics ($p < 0.01$, paired $t$-test).

| PGM type | Model | MSE↓ | VPT-MSE↑ | Dist↓ | VPT-Dist↑ |
|---|---|---|---|---|---|
| Fig. 1C | meta-GRU-res | **1.44(0.34)e-2** | **0.68(0.26)** | **2.88(1.45)** | **0.97(0.07)** |
| | meta-NODE | 1.60(0.26)e-2 | 0.58(0.22) | 6.10(2.63) | 0.80(0.12) |
| | meta-RGN-res | 1.59(0.24)e-2 | 0.56(0.21) | 6.97(3.08) | 0.76(0.13) |
| Fig. 1B2 | GRU-res | 1.63(0.21)e-2 | 0.50(0.17) | 10.4(3.30) | 0.61(0.09) |
| | GRU-res finetune | 1.65(0.24)e-2 | 0.50(0.18) | 9.35(3.33) | 0.66(0.12) |
| | NODE | 1.69(0.18)e-2 | 0.48(0.16) | 10.9(3.32) | 0.59(0.08) |
| | NODE finetune | 1.70(0.19)e-2 | 0.48(0.17) | 10.4(3.23) | 0.61(0.09) |
| | RGN-res | 1.70(0.17)e-2 | 0.47(0.16) | 11.2(3.39) | 0.58(0.09) |
| | RGN-res finetune | 1.72(0.19)e-2 | 0.47(0.17) | 10.0(3.36) | 0.62(0.11) |
| Fig. 1B1 | DVBF | 2.32(14.4)e-2 | 0.02(0.10) | 45.3(0.00) | 0.00(0.00) |
| | DVBF finetune | 2.33(13.4)e-2 | 0.02(0.10) | 45.3(0.00) | 0.00(0.00) |
| | KVAE | 3.37(1.36)e-2 | 0.24(0.19) | 4.81(3.61) | 0.57(0.29) |
| Fig. 1A | meta-DKF | 3.80(0.59)e-2 | 0.10(0.11) | 7.35(3.26) | 0.70(0.25) |
| | DKF | 3.84(0.59)e-2 | 0.10(0.11) | 7.39(3.21) | 0.69(0.25) |
| | DKF finetune | 3.85(0.58)e-2 | 0.10(0.11) | 7.51(3.26) | 0.69(0.25) |
| | VRNN | 1.78(10.9)e-2 | 0.24(0.11) | 23.1(21.6) | 0.51(0.07) |
| | VRNN finetune | 2.15(12.2)e-2 | 0.21(0.16) | 8.31(11.6) | 0.75(0.19) |
| Autoregressive | Donà et al | 3.52(0.26)e-2 | 0.001(0.01) | 13.7(3.05) | 0.06(0.15) |

optimization-based meta-learning of MAML (Finn et al., 2017) to the DKF and GRU-res models, although challenges of stability and convergence as noted in literature (Mehta et al., 2021; Antoniou et al., 2018) were encountered, suggesting that MAML extensions to SLVMs may not be trivial due to issues such as vanishing gradient issues over the complex computation graph.

All GRE-res, NODE, and RGN-res based models were trained to forecast for a sequence of 20 frames using only the first 3 frames. We investigated $k$-shot forecasting when $k$ is fixed at different values of $k = 1, 5, 10, 15$, or allowed to be variable at both meta-training and -test with 15 as the upper limit. For VRNN, DKF, DVBF, and KVAE, we used their public implementations for training and evaluation. Similar network components with the meta-models were scaled to have comparable parameter scales. Because of their reliance on observed time frames to support prediction, 8 observed frames were exposed to the encoder to reconstruct the 8 frames and forecast the additional 12 frames.

**Metrics:** We considered four quantitative metrics on meta-test series. We included the commonly used mean squared error (MSE) of forecasted images, and the recently-proposed metric of Valid Prediction Time (VPT) that measures how long the predicted object's trajectory remains close to the ground truth trajectory based on the MSE (VPT-MSE) (Botev et al., 2021). Because pixel-level MSE does not necessary well capture the quality of the predicted dynamics due to the small object size on the image, we further introduced two new metrics: distance (Dist) between the ground-truth and predicted location of the moving object; and VPT determined based on this distance error (VPT-Dist).

**Comparison with baseline models trained on full dynamics:** For *gravity-16* data, we used 10 gravity in meta-training, 2 in meta-validation, and 4 in meta-testing. Table 1 summarizes the quantitative test performance of the three $k$-shot meta-models obtained with $k = 15$, in comparison to each of the baseline models trained from the full meta-training set. We include complete results across all models in Appendix D with Table 4. Visual examples for these quantitative results are in Appendix D with Fig. 7 (shaded blue): all the baseline models, including their fine-tuned and meta-versions, struggled with limited forecasting ability, especially evidenced by the error in predicting the movement of the ball over time (Dist and VPT-Dist). For DKF/VRNN/KVAE and meta-DKF, there were strong reconstruction and near-term forecasting from partially observed frames (marked by red vertical lines), but incorrect forecasting further away from the observed frames. GRU-res/NODE/RGN-res and their fine-tuned versions exhibited difficulty to describe mixed gravity.

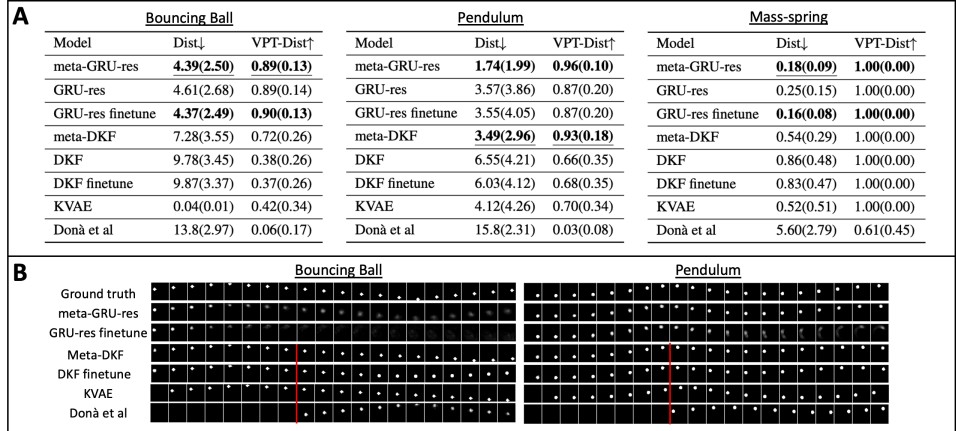

Figure 2: A: Comparison with baselines trained on mixed-physics. B: Forecasting examples.

Table 2: Comparison with baselines trained on single dynamics in meta-training data on gravity-16.

| Model | Dynamics | MSE↓ | VPT-MSE↑ | Dist↓ | VPT-Dist↑ |
|---|---|---|---|---|---|
| meta-GRU-res | known | **1.43(0.34)e-2** | **0.68(0.26)** | **2.86(1.44)** | **0.97(0.06)** |
| | unknown | **1.45(0.33)e-2** | **0.67(0.25)** | **2.96(1.49)** | **0.97(0.07)** |
| GRU-res | known | 1.80(0.29)e-2 | 0.46(0.23) | 6.86(3.95) | 0.77(0.19) |
| | unknown | 1.99(0.277)e-2 | 0.37(0.18) | 8.07(3.56) | 0.69(0.17) |
| GRU-res finetune | unknown | 2.03(0.27)e-2 | 0.35(0.17) | 8.51(3.42) | 0.66(0.18) |
| meta-DKF | known | 3.81(0.59)e-2 | 0.10(0.11) | 7.37(3.27) | 0.70(0.25) |
| | unknown | 3.80(0.59)e-2 | 0.10(0.11) | 7.30(3.21) | 0.70(0.25) |
| DKF | known | 3.74(0.55)e-2 | 0.10(0.11) | 8.37(3.79) | 0.63(0.28) |
| | unknown | 3.79(0.52)e-2 | 0.09(0.10) | 8.72(3.75) | 0.60(0.27) |
| DKF finetune | unknown | 3.82(0.52)e-2 | 0.09(0.10) | 8.77(3.77) | 0.59(0.27) |
| KVAE | known | 3.42(1.30)e-2 | 0.39(0.34) | 5.05(3.57) | 0.5(0.34) |
| | unknown | 3.46(1.36)e-2 | 0.22(0.19) | 5.17(3.91) | 0.53(0.28) |
| Donà et al | known | 3.58(0.33)e-2 | 0.00(0.01) | 13.7(3.36) | 0.07(0.18) |
| | unknown | 3.56(0.34)e-2 | 0.00(0.01) | 14.1(3.84) | 0.08(0.19) |

For *mixed-physics* data, for each of the three physics, we included three dynamic settings in meta-training and left out one in meta-testing. Fig. 2A summarized the test results of the presented meta-GRU-res (with variable $k$) with representative baseline models. Visual examples are shown in Fig. 2B. As shown, meta-DKF, DKF, and DVBF again demonstrated limited ability for long-term forecasting across all physics. KVAE, VRNN, and the finetuned global latent GRU-res were more successful with the mass spring and pendulum systems with relatively simpler dynamics, yet they struggled with the gravity system. The presented meta-GRU-res model consistently outperformed all the baselines across all dynamics, with a larger gain in more complex dynamics.

**Comparison with baseline models trained on single dynamics:** Table 2 summarizes the performance of representative baseline models when trained on a single gravity on *gravity-16* data in comparison to meta-GRU. As shown, in both test dynamics known and unknown to the training, the meta-models outperformed the single-dynamic baselines, suggesting the added benefits of learning across dynamics. This margin of improvements remained even when the single-dynamics baselines were fine-tuned to the $k$-shot support series of unknown test dynamics. Visual examples of these baselines are also shown in Appendix D with Fig. 7 (orange shade).

**Ablation study:** Table 3 summarized the effect of $k$ on $k$-shot forecasting using the meta-GRU-res model. As expected, model performance improved as the size of $k$ increased. Even with $k = 5$, however, the performance was significantly better than all the base models summarized in Table 1.

Table 3: Performance metrics of meta-GRU-res models with fixed vs. variable $k$ values

| K | Mode | MSE↓ | VPT-MSE↑ | Dist↓ | VPT-Dist↑ |
|---|---|---|---|---|---|
| 1 | Fixed | 1.80(0.21)e-2 | 0.44(0.16) | 10.6(3.40) | 0.60(0.10) |
| 5 | Fixed | 1.53(0.36)e-2 | 0.61(0.25) | 3.49(1.89) | 0.94(0.10) |
| 10 | Fixed | 1.46(0.34)e-2 | 0.65(0.26) | 3.08(1.58) | 0.96(0.08) |
| 15 | Fixed | **1.44(0.34)e-2** | **0.68(0.26)** | **2.88(1.45)** | **0.97(0.07)** |
|  | Variable | **1.50(0.34)e-2** | **0.64(0.25)** | **3.44(1.80)** | **0.94(0.10)** |

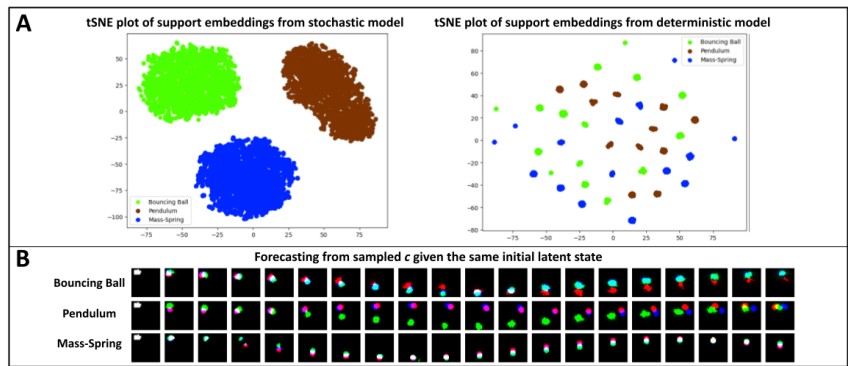

Figure 3: A: t-SNE plot of support-set embedding $\mathbf{c}$ from stochastic (left) and deterministic (right) meta-models. B: Generated forecasting by sampling the distribution of $\mathbf{c}$ given the same $z_0$.

Allowing $k$ to be variable had no noticeable effect on model performance. This flexibility highlights the practicality of the presented framework to forecast with any given size of support series.

**Latent embedding and generation of diverse dynamics:** Fig. 3A shows the distribution of the latent embedding $\mathbf{c}$ obtained from randomly-selected support set, in comparison to a deterministic version of the presented meta-model on *mixed-physics* data. As shown, the presented framework was able to recognize and separate the three dynamics using the $k$-shot support set: given an initial $\mathbf{z}_0$, it was then able to generate different time-series within the same dynamics as well as across dynamics by sampling the distribution of $\mathbf{c}$ ( Fig. 3B). This was not possible with its deterministic counterpart.

# 6 EXPERIMENTS ON COMPLEX PHYSICS SIMULATIONS

We then considered learning and forecasting two more complex physics-based dynamics: turbulent flow dynamics and cardiac electrical dynamics.

**Turbulent flow dynamics:** We customized the meta-GRU model to a dataset of turbulent flow dynamics, simulated with 25 varying buoyant forces acting on the fluid. Each dynamic contains $64 \times 64$ velocity fields of turbulent flows. We use 20 dynamics in meta-training and meta-validation with 80-20 split, and the rest 5 in meta-testing. We followed the experimental setup in (Wang et al., 2021) with an observed window (20 frames in theirs vs. 5 in ours) and a prediction roll-out of 20 frames. Despite using a smaller number of observed frames, the presented meta-GRU model obtained a rooted MSE (RMSE) of $0.26 \pm 0.05$ on seen dynamics and $0.49 \pm 0.05$ on unseen dynamics, in comparison to respective RMSEs of $0.42 \pm 0.01$ and $0.51 \pm 0.02$ reported in (Wang et al., 2021), all reported on the 20 roll-out frames. We included trajectory visualizations in Appendix E.

**Cardiac electrical dynamics:** The propagation of electrical waves in the heart is governed by reaction-diffusion partial differential equations (PDEs) (Aliev & Panfilov, 1996). While direct PDE-based simulation holds clinical potential (*e.g.*, for virtually testing treatment response), its patient-specific parameters are difficult to estimate and its computational cost is high. Although neural approximations provide a promising computationally-efficient alternative (Fresca et al., 2021), how to personalize such a neural model remains an open challenge where existing models are typically trained for a PDE with given parameter configurations. Here, we apply the presented framework

Figure 4: Visual examples (A) and performance metrics of meta-GRU versus other baselines trained on the meta-training set for forecasting electrical dynamics on the heart.

for few-shot learning of a personalized neural model that can be used to efficiently forecast how a patient-specific heart may respond to electrical simulations at different locations.

We simulate electrical propagation originating from various locations in a 3D heart mesh, with 15 settings of PDE parameters representing 15 dynamics with different locations of injury to the heart muscle. We use 9 dynamics in meta-training, 3 in meta-validation, and 3 in meta-testing, with disjoint time-series with different initial conditions (meta-training: 450; meta-test: 2,020). Each time series describes 3D+T propagation of electrical wave with blocks at locations of muscle injury specific to each dynamics (see an example if Fig. 4A column 1). The quality of the forecast series is measured by its MSE and spatial correlation coefficient (CC) with the actual time-series.

We adopted a graph-CNN encoder/decoder and an ODE-GRU latent dynamic function $\mathbf{z}_k = f(\mathbf{z}_{<k}; \boldsymbol{\theta}_z)$ similar to that described in (Jiang et al., 2021). We trained it with a global $\boldsymbol{\theta}_z$ (global GRU), an individual $\boldsymbol{\theta}_z$ for each PDE parameter (single-dynamics GRU), a conditioned $f(\mathbf{z}_{i-1}, \mathbf{c}; \boldsymbol{\theta}_z)$ with $\mathbf{c}$ encoded from individual training series (instance-specific GRU), and the presented framework (meta GRU) with $k$ varying between 1 and 5. We further added a strong personalized virtual heart (PVH) baseline using the original PDE simulation, with the PDE parameter optimized by a SOTA approach from $k$-shot support series as described in (Dhamala et al., 2018).

As shown in Fig. 4A and additional examples in Appendix F, only meta-GRU was able to accurately forecast the propagation block while the other baseline models missed the correct locations of muscle injury specific to a subject (black circles in column 1): note that the single-dynamics GRU performed well on the training dynamics Fig. 4A, but fails on unknown dynamics (Appendix F). Unable to identify injury to patient-specific heart, the forecasting model will be of little value in clinical tasks such as personalized prediction and treatment planning. This gain in forecasting performance by meta-GRU is quantitatively summarized in Fig. 4B across all meta-test time-series. Note that meta-GRU exhibited a notable margin of improvement even versus the PVH: PVH takes on average 5 minutes to forecast each series, versus 0.24 seconds by the meta-GRU; moreover, to optimize PDE parameters of the PVH on average required 100 calls to the PDEs (*i.e.*, $\sim 10$ hours), versus 0.032 seconds for meta-GRU to adapt to patient-specific dynamics. This substantial gain in efficiency without loss of accuracy holds significant value for clinical applications.

## 7 CONCLUSIONS AND DISCUSSION

In this paper, we present a sequential LVM framework to unify existing approaches to learning latent dynamics, identify their limitations associated with the underlying choices of PGMs, and provide empirical evidence for the identified limitations. We further identify meta-learning as an intuitive solution to the identified open gaps, present a framework for few-shot high-dimensional time-series forecasting, and demonstrate that its performance gain is agnostic to the underlying choice of latent dynamic functions. **Limitations:** An avenue of future work is to expand the latent dynamic functions in this framework, especially those integrating strong inductive biases based on physics such as Hamiltonian mechanics (Botev et al., 2021).

## ACKNOWLEDGMENTS

This study was supported by NIH National Heart, Lung, And Blood Institute (NHLBI) grant R01HL145590, NIH National Institute of Nursing Research (NINR) grant R01NR018301, and NSF OAC-2212548.

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

## A  DERIVATION OF EQUATION 7

$$\sum_{j=1}^{M} \sum_{\mathbf{x}_{0:T}^q \in \mathcal{D}_j^q} \log p(\mathbf{x}_{0:T}^q | \mathcal{D}_j^s)$$

$$= \sum_{j=1}^{M} \sum_{\mathbf{x}_{0:T}^q \in \mathcal{D}_j^q} \log \int_{\mathbf{c}} p(\mathbf{x}_{0:T}^q | \mathbf{c}) p_\zeta(\mathbf{c} | \mathcal{D}_j^s) d\mathbf{c}$$

$$= \sum_{j=1}^{M} \sum_{\mathbf{x}_{0:T}^q \in \mathcal{D}_j^q} \log \int_{\mathbf{c}} \int_{\mathbf{z}_0} \left[ p_{\theta_x}(\mathbf{x}_0 | \mathbf{z}_0, \mathbf{c}) \prod_{i=1}^{T} p_{\theta_x}(\mathbf{x}_i | \mathbf{z}_i, \mathbf{c}) \right]$$

$$\cdot \frac{p(\mathbf{z}_0)}{q_\phi(\mathbf{z}_0 | \mathbf{x}_{0:l_{z_0}}^q)} \frac{p_\zeta(\mathbf{c} | \mathcal{D}_j^s)}{q_\zeta(\mathbf{c} | \mathcal{D}_j^s \cup \mathbf{x}_{0:T}^q)} \cdot q_\phi(\mathbf{z}_0 | \mathbf{x}_{0:l_{z_0}}^q) q_\zeta(\mathbf{c} | \mathcal{D}_j^s \cup \mathbf{x}_{0:T}^q) d\mathbf{z}_0 d\mathbf{c}$$

$$\geq \sum_{j=1}^{M} \sum_{\mathbf{x}_{0:T}^q \in \mathcal{D}_j^q} \int_{\mathbf{c}} \int_{\mathbf{z}_0} \left[ \log p_{\theta_x}(\mathbf{x}_{0:T}^q | \mathbf{z}_0^q, \mathbf{c}) - \log \frac{q_\phi(\mathbf{z}_0 | \mathbf{x}_{0:l_{z_0}}^q)}{p(\mathbf{z}_0)} \right.$$

$$\left. - \log \frac{q_\zeta(\mathbf{c} | \mathcal{D}_j^s \cup \mathbf{x}_{0:T}^q)}{p_\zeta(\mathbf{c} | \mathcal{D}_j^s)} \right] \cdot q_\phi(\mathbf{z}_0 | \mathbf{x}_{0:l_{z_0}}^q) q_\zeta(\mathbf{c} | \mathcal{D}_j^s \cup \mathbf{x}_{0:T}^q) d\mathbf{z}_0 d\mathbf{c}$$

$$= \sum_{j=1}^{M} \sum_{\mathbf{x}_{0:T}^q \in \mathcal{D}_j^q} \mathbb{E}_{q_\phi(\mathbf{z}_0^q), q_\zeta(\mathbf{c} | \mathcal{D}_j^s \cup \mathbf{x}_{0:T}^q)} [\log p_{\theta_x}(\mathbf{x}_{0:T}^q | \mathbf{z}_0^q, \mathbf{c})]$$

$$- \mathrm{KL}(q_\phi(\mathbf{z}_0^q | \mathbf{x}_{0:l_{z_0}}^q) || p(\mathbf{z}_0)) - \mathrm{KL}\left( q_\zeta(\mathbf{c} | \mathcal{D}_j^s \cup \mathbf{x}_{0:T}^q) || p_\zeta(\mathbf{c} | \mathcal{D}_j^s) \right)$$

## B  GATED RECURRENT UNIT (GRU) SET-CONDITIONING

We condition the GRU cell of the transition function through Equation 8,

$$\begin{aligned}
\mathbf{z}_{i-1}^{(1)} &= \mathrm{ELU}(\boldsymbol{\alpha}_1 \mathbf{z}_{i-1} + \boldsymbol{\beta}_1 \mathbf{c} + \boldsymbol{\gamma}_1), \quad \mathbf{g}_{i-1} = \sigma(\mathbf{W}_1 \mathbf{z}_{i-1}^{(1)} + \mathbf{b}_1) \\
\mathbf{z}_{i-1}^{(2)} &= \mathrm{ELU}(\boldsymbol{\alpha}_2 \mathbf{z}_{i-1} + \boldsymbol{\beta}_2 \mathbf{c} + \boldsymbol{\gamma}_2), \quad \mathbf{h}_{i-1} = \mathrm{ELU}(\mathbf{W}_2 \mathbf{z}_{i-1}^{(2)} + \mathbf{b}_2) \\
\tilde{\mathbf{z}}_{i-1} &= \boldsymbol{\alpha}_3 \mathbf{z}_{i-1} + \boldsymbol{\beta}_3 \mathbf{c} + \boldsymbol{\gamma}_3, \quad \Delta \mathbf{z}_i = (1 - \mathbf{g}_{i-1}) \odot (\mathbf{W}_3 \tilde{\mathbf{z}}_{i-1} + \mathbf{b}_3) + \mathbf{g}_{i-1} \odot \mathbf{h}_{i-1},
\end{aligned} \tag{8}$$

where $\boldsymbol{\theta}_z = \{\mathbf{W}_i, \mathbf{b}_i, \boldsymbol{\alpha}_i, \boldsymbol{\beta}_i, \boldsymbol{\gamma}_i\}_{i=1}^{3}$ are learnable parameters of the dynamic function.

## C  EXTRA CONSTRAINT TO META-MODEL

We considered adding an additional regularization to the set embedding $p_\zeta(\mathbf{c} | \mathcal{D}_j^s)$ so that it is constrained to a reasonable range. Since the true posterior density $p_\zeta(\mathbf{c} | \mathcal{D}_j^s)$ is unknown, we assume

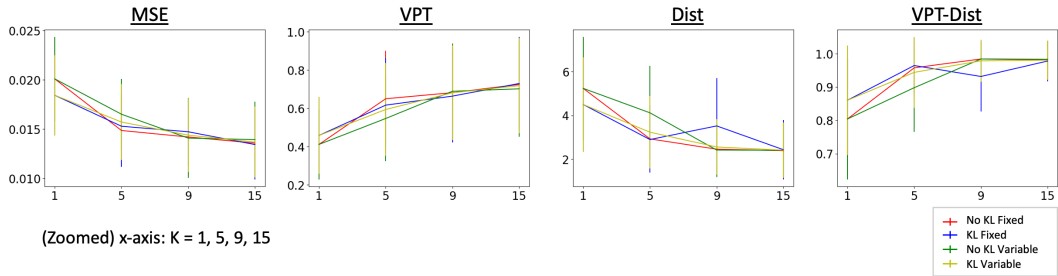

(Zoomed) x-axis: K = 1, 5, 9, 15

Figure 5: Comparison with models with and without constraint.

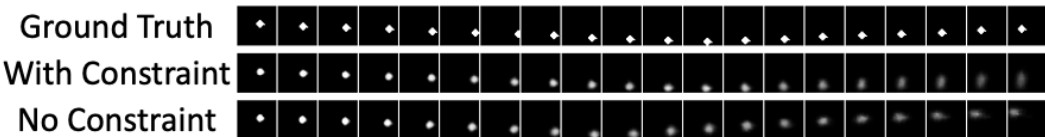

Figure 6: Examples of models with and without constraint.

that it is bounded by a standard Gaussian distribution $\mathcal{N}(0, \mathbf{I})$. Therefore, the objective function of the model becomes:

$$
\arg\min \sum_{j=1}^{M} \sum_{\mathbf{x}_{0:T}^q \in \mathcal{D}_j^q} \mathbb{E}_{q_\phi(\mathbf{z}_0^q), q_\zeta(\mathbf{c} | \mathcal{D}_j^s \cup \mathbf{x}_{0:T}^q)} [\log p_{\theta_x}(\mathbf{x}_{0:T}^q | \mathbf{z}_0^q, \mathbf{c})]
$$
$$
- \mathrm{KL}(q_\phi(\mathbf{z}_0^q | \mathbf{x}_{0:l_{z_0}}^q) || p(\mathbf{z}_0)) - \mathrm{KL}\left(q_\zeta(\mathbf{c} | \mathcal{D}_j^s \cup \mathbf{x}_{0:T}^q) || p_\zeta(\mathbf{c} | \mathcal{D}_j^s)\right)
$$
$$
- \mathrm{KL}\left(p_\zeta(\mathbf{c} | \mathcal{D}_j^s) || \mathcal{N}(0, \mathbf{I})\right)
$$

We applied the extra constraint to the proposed meta-GRU-res model for both fixed and variable $k$ at different value of $k = 1, 5, 9, 15$ and evaluated on *gravity-16* dataset. Fig. 5 summaries the quantitative test performance of the two models trained with and without the constraint on the set embedding. The constraint generally had no noticeable effect on model performance. Visual examples at $k = 1$ are also shown in Fig. 6. It shows that when $k$ is small, the model with the constraint had a slightly better performance.

## D    ADDITIONAL GRAVITY-16 RESULTS

Here we provide the complete results of all baseline models trained on *gravity-16* in Table 4 when split between the known dynamics during training and the unknown ones during testing . Similarly, we provide the full visualization of all baselines within Fig. 7. We note that the VRNN fails to converge in any of the single dynamics cases, in contrast to its full dynamics training, which is likely due to the lower data availability in these settings. The other baselines DKF and KVAE see similar decreases in performance compared to their full meta-training set performance.

## E    TURBULENT FLOW VISUALIZATIONS

Here we provide visualizations of predicted trajectories for both known and unknown buoyancy factors on the turbulent flow dataset within Fig. 8 and Fig. 9.

## F    ADDITIONAL CARDIAC FIGURE

Here we provide an additional result on the cardiac experiments in Figure 10, this time highlighting the performance of the model on unknown dynamic cardiac dynamics when compared against the baselines. We use a varying support set $k$ size between 1 to 5 samples. The proposed model manages

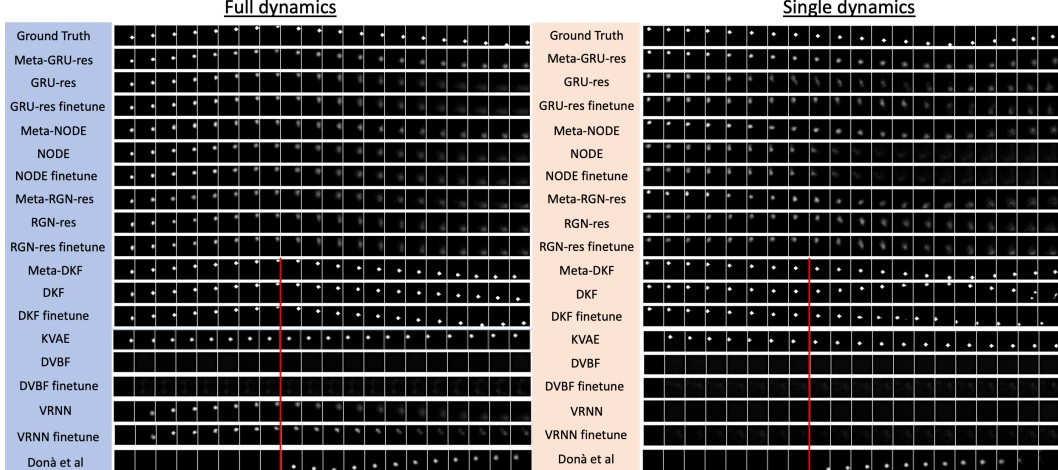

Figure 7: Forecasting on gravity-16 under (blue) full dynamics and (orange) single dynamics training.

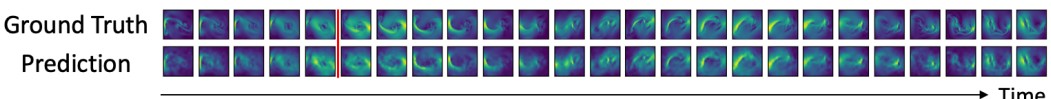

Figure 8: Example of forecasting *known* turbulent flow dynamics.

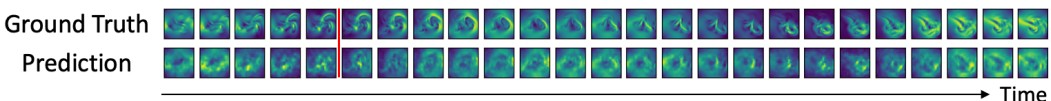

Figure 9: Example of forecasting *unknown* turbulent flow dynamics.

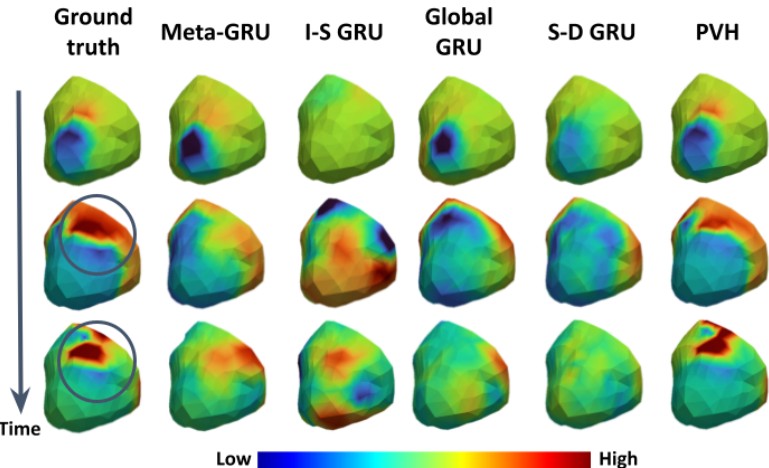

Figure 10: Example of forecasting *unknown* electrical dynamics on the heart.

to effectively model around the scar tissue present in the ground truth. The baselines, besides the expensive personalized virtual heart (PVH), are unable to account for the specific dynamics of this subject and propagate over it.

Table 4: Comparison of the presented meta and base models on known and unknown dynamics.

| Model | Dynamics | MSE↓ | VPT-MSE↑ | Dist↓ | VPT-Dist↑ |
|---|---|---|---|---|---|
| meta-GRU-res | known | **1.43(0.34)e-2** | **0.68(0.26)** | **2.86(1.44)** | **0.97(0.06)** |
| | unknown | **1.45(0.33)e-2** | **0.67(0.25)** | **2.96(1.49)** | **0.97(0.07)** |
| meta-NODE | known | 1.60(0.26)e-2 | 0.58(0.23) | 6.05(2.65) | 0.80(0.12) |
| | unknown | 1.62(0.26)e-2 | 0.57(0.22) | 6.23(2.54) | 0.80(0.12) |
| meta-RGN-res | known | 1.59(0.25)e-2 | 0.57(0.21) | 6.89(3.08) | 0.76(0.13) |
| | unknown | 1.60(0.23)e-2 | 0.56(0.20) | 7.23(3.09) | 0.75(0.13) |
| GRU-res | known | 1.80(0.29)e-2 | 0.46(0.23) | 6.86(3.95) | 0.77(0.19) |
| | unknown | 1.99(0.277)e-2 | 0.37(0.18) | 8.07(3.56) | 0.69(0.17) |
| GRU-res finetune | unknown | 2.03(0.27)e-2 | 0.35(0.17) | 8.51(3.42) | 0.66(0.18) |
| NODE | known | 1.97(0.21)e-2 | 0.30(0.20) | 10.6(4.13) | 0.60(0.17) |
| | unknown | 2.06(0.22)e-2 | 0.27(0.17) | 11.1(3.71) | 0.57(0.16) |
| NODE finetune | unknown | 2.07(0.21)e-2 | 0.27(0.17) | 11.2(3.72) | 0.55(0.16) |
| RGN-res | known | 1.93(0.22)e-2 | 0.34(0.20) | 10.1(4.11) | 0.61(0.18) |
| | unknown | 2.03(0.22)e-2 | 0.30(0.16) | 10.8(3.72) | 0.57(0.16) |
| RGN-res finetune | unknown | 2.05(0.22)e-2 | 0.29(0.16) | 10.7(3.62) | 0.56(0.16) |
| meta-DKF | known | 3.81(0.59)e-2 | 0.10(0.11) | 7.37(3.27) | 0.70(0.25) |
| | unknown | 3.80(0.59)e-2 | 0.10(0.11) | 7.30(3.21) | 0.70(0.25) |
| DKF | known | 3.89(0.32)e-2 | 0.08(0.06) | 10.7(3.17) | 0.46(0.20) |
| | unknown | 3.88(0.32)e-2 | 0.08(0.06) | 10.7(3.23) | 0.45(0.20) |
| DKF finetune | unknown | 3.89(0.32)e-2 | 0.08(0.06) | 10.8(3.24) | 0.45(0.20) |
| VRNN | known | 2.32(14.6)e-2 | 0.02(0.11) | 45.3(0.00) | 0.00(0.00) |
| | unknown | 2.32(14.6)e-2 | 0.01(0.08) | 45.3(0.00) | 0.00(0.00) |
| VRNN finetune | unknown | 2.34(14.5)e-2 | 0.04(0.15) | 45.3(0.00) | 0.00(0.00) |
| DVBF | known | 2.32(14.3)e-2 | 0.01(0.02) | 45.2(0.00) | 0.00(0.00) |
| | unknown | 2.43(14.1)e-2 | 0.01(0.07) | 45.3(0.00) | 0.00(0.00) |
| DVBF finetune | unknown | 2.35(14.1)e-2 | 0.01(0.08) | 45.3(0.00) | 0.00(0.00) |
| KVAE | known | 3.42(1.30)e-2 | 0.39(0.34) | 5.05(3.57) | 0.5(0.34) |
| | unknown | 3.46(1.36)e-2 | 0.22(0.19) | 5.17(3.91) | 0.53(0.28) |
| Donà et al | known | 3.58(0.33)e-2 | 0.00(0.01) | 13.7(3.36) | 0.07(0.18) |
| | unknown | 3.56(0.34)e-2 | 0.00(0.01) | 14.1(3.84) | 0.08(0.19) |

# G  DATA DETAILS

In this section, we give the specific data sizes and splits used for training throughout the experiments, as well as generation procedures and considerations for each.

For the bouncing balls, we leveraged the PyMunk Physics Engine (www.pymunk.org) to perform simulations under various gravity following (Fraccaro et al., 2017). For pendulum and mass spring systems, we leveraged the Hamiltonian Dynamics Suite presented in (Botev et al., 2021). The suite's default physical parameters were used and friction coefficients were introduced to build non-energy conserving systems. For data consistency, we extracted the red color channel of pendulum and mass spring systems to generate gray-scale images.

For *gravity-16* experiments, we generated bouncing balls dynamic trajectories on $32 \times 32$ images. All dynamics are under 16 different gravitational constants, where $g = 3 + \epsilon, \epsilon \sim U(0, 1)$ and the direction of each gravity is evenly distributed over the 2D space. We took the trajectories samples at every $\Delta t = 0.2$ intervals. The initial position of the ball is set to a $16 \times 16$ region centered in the image, and the initial velocity is randomly sampled from $[0, 10]$ on both $x$ and $y$ directions. Each gravity setting has 3,000 samples in total. For *gravity-16* data, we used 10 gravity in meta-training, 2 in meta-validation, and 4 in meta-testing. We also left out samples from both meta-training and

meta-validation sets ($\sim$1500 for each gravity) to evaluate the performance of the model on the known dynamics.

For *mixed-physics* experiments, we generated a mixed-physics dataset consisting of bouncing balls under 4 gravity directions, and pendulums and mass springs each with four different values of friction coefficients of $0, 0.05, 0.1, 0.15$. The bouncing ball dataset is similar to *gravity-16* experiment except for the 4 gravity directions. Both pendulums and mass springs took the trajectories samples at every $\Delta t = 0.2$ intervals. In pendulums, the mass of the particle $m = 0.5$, the gravitational constant $g = 3$ and the pivot length $l = 1$. The friction coefficients are chosen in $0, 0.05, 0.1, 0.15$. In mass springs, the mass of the particle $m = 0.5$, the spring force coefficient $k = 2$. The friction coefficients are chosen in $0, 0.05, 0.1, 0.15$. For each of the three physics, we included three dynamic settings in meta-training and left out one in meta-testing.

For the turbulent flow dataset, we generated the tasks as given by the instructions and scripts found within the official code repository from (Wang et al., 2021), `https://github.com/Rose-STL-Lab/Dynamic-Adaptation-Network`. We split the seen and unseen buoyancy factors according to the same task split used within their work and directly compare RMSE values based on their implementation and magnitude coefficients.

For cardiac electrical dynamics, we generated 3D electrical signal propagation in the heart simulated by the Aliev-Panfilov model Aliev & Panfilov (1996) on 3 heart meshes and a total of 12 different tissue parameters (4 on each heart) representing different injury to the heart muscle. This was treated as 12 tasks in meta-learning. All 12 tasks appeared in meta-training and -testing, with disjoint time sequences resulting from different external stimulations (meta-training: 300; meta-test: 2,020).

All ball data can be found here: `https://drive.google.com/drive/folders/1Tm3DNrugcSbWXSNyeGL3jQKR8y3iXx0m?usp=sharing`. The heart data can be found here: `https://drive.google.com/drive/folders/12S579V0KWMgbHGXDQZt0rQyfzF1AyNCu?usp=sharing`.

## H  IMPLEMENTATION DETAILS

In this section, we give the specific hyper-parameters on each experiment over all models, as well as resources and considerations for each. All experiments were run on NVIDIA Tesla T4s with 16 GB memory.

### H.0.1  ARCHITECTURE FOR META MODELS

The implementation of our proposed meta models is here: `https://github.com/john-x-jiang/meta_ssm`. Specifically in the configuration file, the type of transition function can be changed by `trans_mode` and `trans_args` in the model section. The size of the support set is controlled by `k_shot` in the data section, and the variable/fixed $K$ is set up by `changeable` in both training and evaluation section. During meta-testing, the paired support and query sets should be put under `eval_tags` and `pred_tags` in the data section. Detailed hyperparameter settings are shown below.

**Meta Model Architecture on Mixed-Physics and Gravity-16**

- Domain Input: 20 observation timesteps of $32 \times 32$ dimensions
- Initialization Input: 3 observation timesteps of $32 \times 32$ dimensions
- Optimizer: Adam, $5 \times 10^{-4}$ learning rate
- Transition: Gated transition function (GRU-res) / Recurrent Generative Network (RGN-res) / Neural Ordinary Differential Equation (NODE)
- Batch size: 50
- Number of epochs: 200
- Training time: 1.5 - 5 hours
- Latent Units: 8

- Transition Units: 100
- Domain Encoder Filters: [8, 16, 8]
- Domain Time Units: [10, 5, 1]
- Initial Encoder Filters: [8, 16, 8]
- Emission Filters: [32, 16, 8, 1]
- KL term initialization: $\lambda_1 = 10^{-2}$
- KL term set-embedding: $\lambda_2 = 10^{-2}$

**Meta Model Architecture on Turbulent Flow**

- Domain Input: 20 observation timesteps of $64 \times 64$ dimensions
- Initialization Input: 5 observation timesteps of $64 \times 64$ dimensions
- Optimizer: Adam, $5 \times 10^{-4}$ learning rate
- Transition: Gated transition function (GRU-res) / Recurrent Generative Network (RGN-res) / Neural Ordinary Differential Equation (NODE)
- Batch size: 20
- Number of epochs: 500
- Training time: 6 hours
- Latent Units: 64
- Transition Units: 100
- Domain Encoder Filters: [32, 64, 128, 32]
- Domain Time Units: [10, 5, 2, 1]
- Initial Encoder Filters: [64, 128, 256, 64]
- Emission Filters: [256, 128, 64, 1]
- KL term initialization: $\lambda_1 = 10^{-1}$
- KL term set-embedding: $\lambda_2 = 10^{-1}$

### H.0.2 ARCHITECTURE FOR BASELINE MODELS

The implementation of baseline models is here: `https://github.com/john-x-jiang/meta_ssm`. The setting of transition function can be changed by `trans_mode` and `trans_args` in the model section. The testing sets should be put under `pred_tags` in the data section. Detailed hyperparameter settings are shown below.

**Baseline Model Architecture**

- Initialization Input: 3 observation timesteps of $32 \times 32$ dimensions
- Optimizer: Adam, $5 \times 10^{-4}$ learning rate
- Transition: Gated transition function(GRU-res) / Recurrent Generative Network (RGN-res) / Neural Ordinary Differential Equation (NODE)
- Batch size: 50
- Number of epochs: 200
- Training time: 1.3 hours
- Latent Units: 8
- Transition Units: 100
- Initial Encoder Filters: [8, 16, 8]
- Emission Filters: [32, 16, 8, 1]
- KL term initialization: $\lambda_1 = 10^{-2}$

### H.0.3 ARCHITECTURE FOR DKF

The DKF model (Krishnan et al., 2017) we used is based on this implementation: `https://github.com/yjlolo/pytorch-deep-markov-model`. Our implementation is provided in our code. We modified the script to perform reconstruction of the observed part of input sequences and prediction of the unobserved parts. Specifically, the output of the correction function was used in the decoder in the reconstruction phase, while the transition module was used for prediction. Detailed hyperparameter settings are shown below.

**DKF Architecture**

- Input: 8 observation and 12 prediction timesteps of $32 \times 32$ dimensions
- Optimizer: Adam, $5 \times 10^{-4}$ learning rate
- Transition: Gated transition function
- Batch size: 50
- Number of epochs: 200
- Training time: 1.2 hours
- Encoder Units: [2048, 2048, 100]
- RNN Units: 100
- Correction Units: 100
- Transition Units: 100
- Emission Filters: [32, 16, 8, 1]

### H.0.4 ARCHITECTURE FOR DVBF

The PyTorch implementation of DVBF (Karl et al., 2017) we used can be found here: `https://github.com/gregorsemmler/pytorch-dvbf`. Our implementation is provided in our code. We modified this script to perform initial state generation, as highlighted in DVBF's experiments (Karl et al., 2017). We found lacking convergence in more complex dynamics and lower data-availability scenarios, specifically bouncing ball settings. We detail the hyperparameter values chosen per experiment with respect to the hyperparameters found in this repository.

**DVBF Architecture on Gravity-16**

- Input: 8 observation and 12 prediction timesteps of $32 \times 32$ dimensions
- Optimizer: Adam, $1 \times 10^{-3}$ learning rate annealed to $1 \times 10^{-4}$ over 100 epochs
- Delayed KL Weight: between epochs=[5,25], linearly annealed from 0.01 to 1
- Batch-size: 25
- Number of Epochs: 100
- Training time: 15 hours
- Number of matrices $\alpha_t$: 8
- Code vector dim $w_t$: 8
- Feature dim $z_t$: 8
- State dim $s_t$: 1024
- RNN Hidden Size: 100
- Inference Size: 32
- Transition Size: 64
- Encoder CNN Filters: [8, 32, 8]
- Decoder CNN Filters: [8, 32, 8]

**DVBF Architecture on Mixed-Physics**

- Input: 8 observation and 12 prediction timesteps of $32 \times 32$ dimensions
- Optimizer: Adam, $1 \times 10^{-3}$ learning rate annealed to $1 \times 10^{-4}$ over 100 epochs
- Delayed KL Weight: between epochs=[5,25], linearly annealed from 0.01 to 1
- Batch-size: 25
- Number of Epochs: 100
- Training time: 15 hours
- Number of matrices $\alpha_t$: 8
- Code vector dim $w_t$: 8
- Feature dim $z_t$: 8
- State dim $s_t$: 1024
- RNN Hidden Size: 100
- Inference Size: 32
- Transition Size: 64
- Encoder CNN Filters: [8, 32, 8]
- Decoder CNN Filters: [8, 32, 8]

### H.0.5 ARCHITECTURE FOR KVAE

The public implementation of KVAE (Fraccaro et al., 2017) we used can be found here: `https://github.com/simonkamronn/kvae`. No significant modifications were made to the repo - primarily just output formatting. When attempting to scale the hyperparameter K (number of mixture components for the LG-SSM matrices), we ran into numerical loss instability and lacking convergence in reconstruction. We detail the hyperparameter values chosen per experiment with respect to the hyperparameters found in this repository.

**KVAE Architecture on Gravity-16**

- Input: 8 observation and 12 prediction timesteps of $32 \times 32$ dimensions
- Optimizer: Adam, $7 \times 10^{-3}$ learning rate exponentially decayed by 0.85 over 20 epochs
- Batch-size: 32
- Number of Epochs: 80
- Training time: 2 hours
- K: 3
- Latent variable dim $a_t$: 4
- Latent state dim $z_t$: 8
- Control dim $u_t$ (unused): 1
- RNN Hidden Size: 50
- RNN Layers: 2
- Encoder CNN Filters: [3, 32, 32, 32]
- Decoder CNN Filters: [3, 32, 32, 32]

**KVAE Architecture on Mixed-Physics**

- Input: 8 observation and 12 prediction timesteps of $32 \times 32$ dimensions
- Optimizer: Adam, $7 \times 10^{-3}$ learning rate exponentially decayed by 0.85 over 20 epochs
- Batch-size: 32
- Number of Epochs: 80

- Training time: 2 hours
- K: 3
- Latent variable dim $a_t$: 4
- Latent state dim $z_t$: 8
- Control dim $u_t$ (unused): 1
- RNN Hidden Size: 50
- RNN Layers: 2
- Encoder CNN Filters: [3, 32, 32, 32]
- Decoder CNN Filters: [3, 32, 32, 32]

### H.0.6 ARCHITECTURE FOR VRNN

The DVAE code implementation Girin et al. (2021) of VRNN (Chung et al., 2015) we used can be found here: `https://github.com/XiaoyuBIE1994/DVAE/blob/master/dvae/model/vrnn.py`. The model was modified to take its own reconstructions as inputs to the feature extractor at every timestep after the true sequence observation period. We detail the hyperparameter values chosen per experiment with respect to the hyperparameters found in this repository.

**VRNN Architecture on Gravity-16**

- Input: 8 observation and 12 prediction timesteps of $32 \times 32$ dimensions
- Optimizer: Adam, $1 \times 10^{-3}$ learning rate
- Batch-size: 50
- Number of Epochs: 80
- Training time: 8 hours
- Latent state dim $z_t$: 32
- Dropout probability: 0.2
- Dense X Size: 128
- Dense Z Size: 128
- Dense H(X)-Z Size: 128
- Dense H(Z)-X Size: 128
- Dense H(Z) Size: 128
- RNN Layers: 2
- RNN Dim: 64
- Beta coefficient: 1.0
- Activation: LeakyReLU(1.0)

**VRNN Architecture on Mixed-Physics**

- Input: 8 observation and 12 prediction timesteps of $32 \times 32$ dimensions
- Optimizer: Adam, $1 \times 10^{-3}$ learning rate
- Batch-size: 50
- Number of Epochs: 80
- Training time: 8 hours
- Latent state dim $z_t$: 32
- Dropout probability: 0.2
- Dense X Size: 128
- Dense Z Size: 128

- Dense H(X)-Z Size: 128
- Dense H(Z)-X Size: 128
- Dense H(Z) Size: 128
- RNN Layers: 2
- RNN Dim: 64
- Beta coefficient: 1.0
- Activation: LeakyReLU(1.0)

