# OpenReview forum: "Sequential Latent Variable Models for Few-Shot High-Dimensional Time-Series Forecasting"
_ICLR.cc/2023/Conference — ICLR 2023 notable top 25%_

### Official Review · Reviewer_wdac · 2022-10-24

**Confidence:** 3
**Correctness:** 3
**Technical Novelty And Significance:** 2
**Empirical Novelty And Significance:** 2
**Recommendation:** 6

**Clarity, Quality, Novelty And Reproducibility:**

Novelty:

The problem setting and application of sequential latent variable models to few-shot forecasting is super interesting. The overall structure of the presentation and writing is easy to follow.


Quality and Clarity:

Many details of the writing are not satisfying, including the following concerns:
1. In Equation 4 and Equation 7, density functions of p(c) and p($\theta$) are missing in the integrand.
2. Experiments are conducted on forecasting. Nowhere in the paper shows how the forecasting results are obtained. Are the forecasting results obtained by sampling from the variational posterior distribution?
3. Section 3 claims two unifying frameworks for existing sequential latent variable models. For the sake of clarity and self-containedness, the author should consider including, at least in the supplementary material, examples of how existing sequential LVMs can be interpreted under the proposed unifying framework.


I have no concerns about the reproducibility of the experiment results.


**Strength And Weaknesses:**

Strengths:

The problem setting of few-short forecasting for time-series data is important. I also appreciate the motivation of using latent representation for modeling high-dimensional time-series data. The set of models being considered in the experiments is also extensive.


Weakness:

1. Despite considering an extensive set of models, the experiments are not thorough and comparisons are not fair. The training and evaluation settings of the proposed model are essentially meta-learning settings. Why popular meta-learning frameworks like MAML [1] and Probablistic MAML[2] are not considered? I do not think there’re big challenges to extending MAML frameworks to sequential LVMs. For the proposed framework, both the evaluation setting and training settings are episodic where the support set is fed to the model in each training episode. For the two baseline frameworks being compared, the support set is not present during training.
2. Despite the images themselves being high-dimensional, the actual dynamics that generate the image sequences are all quite simple and low-dimensional. I’m not fully convinced the set of experiments shows the model’s capacity to handle real high-dimensional sequential data. As long as the model could uncover the simple low-dimensional ground-truth dynamics from the support set dynamics and generate images of high quality, it should perform well.
3. The technical contribution of the work is incremental. It claims it proposes a framework for few-shot forecasting for sequential data but I do not see significant differences between the proposed framework and existing sequential latent variable models. More specifically, I think the difference between embedding c and latent variable z is only nominal with c being viewed as a conditional latent variable. Actually, the model’s code in the supplementary material also shows no significant difference. This is not necessarily bad. However, without extensive and convincing experiment results, I think the contribution of simply extending existing methods to a new problem setting is very limited.
4. I also have some concerns about the quality writing and clarity of the paper. Please refer to Clarity, Quality, and Novelty.


[1] Finn, Chelsea, Pieter Abbeel, and Sergey Levine. "Model-agnostic meta-learning for fast adaptation of deep networks." International conference on machine learning. PMLR, 2017.

[2] Finn, Chelsea, Kelvin Xu, and Sergey Levine. "Probabilistic model-agnostic meta-learning." Advances in neural information processing systems 31 (2018).

**Summary Of The Paper:**

The work proposes a framework for sequential latent variable models for high-dimensional data few-shot forecasting problems. The model encodes the support set of few-short samples into a distribution of parameters of met -models. The model is optimized in a VAE framework and used for forecasting tasks. Experiment results on image sequence data simulated using diverse dynamics show strong performance of the model against baselines.

**Summary Of The Review:**

Few-shot forecasting is an interesting application for existing sequential latent variable models. However, the experiment settings are flawed and comparisons against baseline approaches are not fair. Important meta-learning baselines including MAML are missing from the experiment results. There are also minor concerns about the writing quality of the paper.

---

> ### Author Response · Authors · 2022-11-19
> **Response to Reviewer wdac**
>
> **Fundamental difference between modeling $q(c)$ and $q(z_k)$***
>
> Because the reviewer’s comment that “the difference between embedding $c$ and latent variable $z$ is only nominal” (bullet 3 in Weakness) touches the fundamental premise of the presented work, we feel it necessary to clarify first to set the context of the rest of our response.
>
> There is a fundamental difference between $c$ and $z_k$: $z_k$ is time-varying and represents the latent state of observed $x_k$ at each time frame $k$, while $c$ is a time-invariant embedding that explains the generation of the entire sequence $x_{0:T}$.
>
> For the former, the prediction of $z_{k+1}$ and $x_{k+1}$ relies on the inference of $q(z_k |x_{\leq k}, z_{<k})$ from observations $x_k$ up to time-frame $k$ – it is thus good for reconstructing a time sequence, but the latent dynamic model is limited to forecast without near-term $x$’s (as it is not trained to do so). For the latter, $q(c)$ explains the property of the dynamic system and, once inferred (from context examples in our work), $x_{0:T}$ can be forecasted given only several initial frames.
>
> In classic state-space systems, these two correspond to two very different and well-recognized concepts: “state estimation” (the former) vs. “system identification” (the latter). The former is designed for uncovering the latent states underlying observed time sequences. The latter is for identifying system parameters for forecasting.
>
> This fundamental difference is the main point of our work (as summarized in Fig 1). Thus we use this opportunity to provide further evidence, using both the vanilla models and their meta-extensions representing these two formulations. For simplicity, from here on we refer to the former $q(z_k)$ based formulation (Fig. 1A) as state-estimation-models, and the latter $q(c)$ based formulation (Fig. 1C) as parameter-id-models.
>
> **Vanilla models:** Performance of the base DKF (representing state-estimation-model) and GRU-res (representing parameter-id-models) are already provided in the original submission and we highlight here (Fig. 2B in main text, and Fig. 7 in Appendix): DKF is excellent at reconstructing observed time frames, but limited in forecasting ahead without near-term observations. GRE-res is good at long-term forecasting when modeling one dynamics (Table 2), but has trouble to model multiple dynamics at the same time (Table 1).
>
> **Feed-forward Meta-models:** The presented framework (Fig 1C) represents a feed-forward meta-model on system-id-models. As suggested by reviewer EkWh, we added a feed-forward meta-extension to the DKF (state-estimation-models) – this can also be seen as an extension of the Sequential Neural Process (SNP) to time-sequence forecasting. Please refer to bullet 2 of our overall response (and the revised manuscript for more details). Overall, these results 1) provide further evidence of the differences among state-estimation vs. system-id models, and 2) add further evidence for the improved performance of the presented meta-model in forecasting diverse dynamics.
>
>
> **Optimization-based meta-models (MAML extension):** As suggested by the reviewer, we also experimented with MAML extension to DKF and GRU-res. While the extension of the MAML framework was relatively straightforward (as anticipated by the reviewer), we were not able to obtain satisfactory results in either state-estimation or system-id model. Please see bullet 3 of our overall response for detailed descriptions.
>
> For completeness, we report the metrics of the two MAML extensions applied to the *gravity-16* and *mixed-physics* datasets below. In the revised manuscript, we omitted the MAML baseline results but instead added a discussion noting the aforementioned challenges in attempting MAML.
>
> | Gravity-16 | MSE &#8595; | VPT &#8593; | Dist &#8595; | VPT-Dist &#8593; |
> |---|---|---|---|---|
> MAML-GRU-res         | 3.06(2.64)e-02 | 0.01(0.02) | 18.2(5.63) | 0.10(0.13) |
> MAML-DKF                | 2.78(2.34)e-02 | 0.01(0.03) | 14.1(4.98) | 0.12(0.21) |
>
> | Mixed-Physics | MSE &#8595; | VPT &#8593; | Dist &#8595; | VPT-Dist &#8593; |
> |---|---|---|---|---|
> MAML-GRU-res        | 2.48(2.79)e-02 | 0.17(0.35) | 8.93(6.50) | 0.41(0.38) |
> MAML-DKF               | 2.75(3.57)e-02 | 0.05(0.18) | 14.0(8.46) | 0.30(0.43) |
>
> Seeing that a straightforward meta-formulation of sequential LVMs does not exist, we hope that this convinces the reviewer that the presented meta-learning work is by no means incremental.

---

> > ### Author Response · Authors · 2022-11-19
> > **Response to Reviewer wdac**
> >
> > **Additional experiments on more complex dynamics**
> >
> > We would like to first note that the gravity-16 and mixed-physics datasets are benchmark data commonly used in related works and we further extended them to a level of diversity beyond those usually reported in existing works. We also would like to note that the cardiac application represents complex underlying dynamics that are by no means simpler than those typically reported in published works. We thus respectfully disagree with the reviewer’s comment regarding “simple low-dimensional ground-truth dynamics”.
> >
> > However, we did add another dataset that was used in a most recent 2022 NeurIPS publication and demonstrated the improved performance of our presented model over the newly-published method. Please refer to bullet 4 of our overall response for more details.
> >
> > **Clarity**
> >
> > - We added $p(\theta)$ In Equation 4 (now Equation 3) in the integrand – thanks for catching this typo. Equation 7 (now Equation 6) does not need $p(c)$ in the integrand as it describes a conditional density $p(x|c)$.
> > - The forecasting results are obtained by 1) sampling from an initial state $q(z0|x_{0:l})$, 2) sampling from the context embedding from few-shot context samples $q(c|D)$, and 3) propagating the sequence forward to generate $z_{1:T}$ via the dynamic transition model and $x_{1:T}$ via the emission model. In section II of the original manuscript, we described the base forecasting model: “a set of models (Fig. 1B2) have been presented that aims to learn a latent dynamic function that forecasts a sequence using only an inferred initial state”. This was then mathematically described in Equation (3) for the base model and in Equation (6) for the meta-model, and graphically presented in Fig 1. In Experiments, we further wrote that “All of these models were trained to forecast for a sequence of 20 frames using only the first 3 frames.”.  To make this furthermore clear, in the revised manuscript we added a summary at the end of section 4: “For each query series,  starting with an initial latent state $z_0$ (inferred from $l_{z_0}$ frames) and k-shot support embedding $c$, the latent dynamic function is asked to propagate forward to forecast the entire sequence of $z_{0:T}$ and their corresponding high-dimensional observations  $x_{0:T}$”
> > - We considered the unifying framework as a main contribution and premise of this paper. Therefore, throughout the Introduction (paragraphs 3-4), Related works (section of “Sequential LVMs”), and Section 3 (in the discussion of each formulation), we have linked specific existing sequential LVMs to each formulation. Specifically, in Section 3, we have dived into how different existing sequential LVMs can be interpreted under the corresponding formulation (see texts under Equation (2) and (4)). This was a key focus we elaborated on in the original submission (providing unifying formulations to link existing representative LVMs), and here we draw the reviewer’s attention to these sections again.

---

> > > ### Author Response · Authors · 2022-11-29
> > > **Reminder to Reviewer wdac**
> > >
> > > Dear Reviewer wdac
> > >
> > > Thank you again for your constructive feedback. We hope that you have had time to go through our response and revised manuscript in addressing your previous comments. We would like to follow up to see if you have additional comments for further discussion.
> > >
> > > Best regards,
> > >
> > > Authors

---

> > ### Comment · Reviewer_wdac · 2022-12-08
> > **Update after Reviews**
> >
> > Thanks for the detailed response. I'm satisfied with the result and updated the score.

---

### Official Review · Reviewer_EkYh · 2022-10-25

**Confidence:** 3
**Correctness:** 4
**Technical Novelty And Significance:** 3
**Empirical Novelty And Significance:** 3
**Recommendation:** 6

**Clarity, Quality, Novelty And Reproducibility:**

The paper is generally well written. The presentation is generally clear, and references are adequately cited.

**Strength And Weaknesses:**

Strengths:
- The dynamic setting has received less attention for meta-learning compared to the iid case, so this work fills a gap in the current literature.
- Empirical results suggest that the proposed method outperforms previous benchmark models significantly for a variety of applications/data sets.

Negatives:
- I feel that the approach is somewhat incremental from a methodology perspective in that it is an extension from previous work in few-shot applied to the iid setting (e.g., Garnelo et al., 2018), in combination with variational inference techniques for sequential latent variables.
- It is not clear to me how is this different to the sequential neural process setting and why it cannot be included here as a comparison? The other benchmark models presented in this work are similarly not designed for a few-shot setup.

Comments/Actionable Feedback:
- How exactly is the variational approximation of c given by the query and support sets constructed? Are the embedded features of the query set just incorporated into the averaging function? Is there some weighting of the query vs support set?

**Summary Of The Paper:**

The authors suggest few-shot learning of sequential latent variable models. The paper considers an amortized variational inference setup. This is obtained by an average-set-pooling/encoding of the support set to learn a context variable for each time series that drives the deterministic dynamics of the latent states. The method is illustrated for meta-learning image sequences and cardic electrical dynamics, where it performs favourable compared to sequential latent variable models that are not trained based on a meta-learning approach.

---
Update post-rebuttal:
I increased my score to a weak accept after reading the authors' response and the other reviews. However, I still think this is a borderline case.
The additional experiments provide empirical evidence that a meta-learning setting of DKF following the Sequential Neural Process (SNP) approach yields inferior forecasting performance compared to the suggested method.

---

**Summary Of The Review:**

The paper addresses few-shot learning for sequential latent variable models. My main concern is that it is not clear to me how this improves on sequential neural processes. For the latter, the context observations can even by varying through time, and do not have to be the support set? There could be a misunderstanding on my side, and I will consider increasing my score if the authors address these concerns.

---

> ### Author Response · Authors · 2022-11-19
> **Response to Reviewer EkYh**
>
> **Relation and comparison with Sequential NP**
>
> The original SNP is different from the presented work in two main aspects.
> - First, SNP is formulated for learning a “supervised” regression function $y = f(x)$ where both x and y are defined over time. Example regression functions in the SNP literature include learning image intensity $y_t$ as a function of pixel locations $x_t$ in a sequence of image completion tasks (part of the image available at each frame), or the images $y_t$ as a function of the camera viewpoints $x_t$ in modeling dynamic 3D scenes. In other words, at any time frame $t$, we need input $x_t$ to predict the output $y_t$ of interest. This is why SNP is not directly suitable for (and has not been used for) forecasting formulation.
> - Second, the LVM in SNP is based on the time-varying latent state $z_t$, similar to the formulation outlined in Fig 1A in our paper. This means that the latent dynamic model relies on the encoder to embed $q(z_t)$ in order to predict $p(z_{t+1})$, weakening its ability to forecast without near-term observations. Similarly, as pointed out by the reviewer, the context variable $c_t$ is also time-varying.
> - Although we cannot directly apply the original implementation of the published SNP to the forecasting problem, we did take this opportunity to design and implement an extension of SNP to obtain a meta-formulation of the baseline model (i.e., the DKF) representing Fig1A. More specifically, while the DKF has an inference model of $q(z_t | x_t,z_{< t})$ and a transition model $p(z_t | z_{<t})$, following the spirit of the SNP, we 1) use a feedforward meta-model to embed the support series ${x_{t,i}}$, $i$ in support set to $c_t$; and 2) condition both $p(z_t)$ and $q(z_t)$ above on $c_t$. Both 1) and 2) follow the formulations presented in the SNP. The meta-model follows the same architecture as what is used in the presented model (except $c_t$ is time-varying), and the model is trained using exactly the same meta-training set and episodic training procedure. This meta-formulation can thus be seen as a counterpart of the presented meta-formulation to the LVM model in Fig 1A.
>
> The results below show that the meta-formulation, which slightly improves the base formulation of DKF, does not overcome the fundamental limitation of this type of LVM outlined in Fig. 1A: because of the reliance of the transition function on the inference latent state $q(z_k)$, this formulation – while strong at reconstruction – is limited in long-term forecasting without near-term observations to support $q(z_k)$. The use of support embedding $c_k$, as shown in the results, does not overcome this limitation.
>
> | Gravity-16 | MSE &#8595; | VPT &#8593; | Dist &#8595; | VPT-Dist &#8593; |
> |---|---|---|---|---|
> DKF                   | 3.84(0.59)e-2 | 0.10(0.11) | 7.39(3.21) | 0.69(0.25)
> meta-DKF          | 3.80(0.59)e-2 | 0.10(0.11) | 7.35(3.26) | 0.70(0.25)
>
> | Mixed-Physics | MSE &#8595; | VPT &#8593; | Dist &#8595; | VPT-Dist &#8593; |
> |---|---|---|---|---|
> DKF                   | 2.81(1.52)e-2 | 0.35(0.38) | 5.73(4.85) | 0.68(0.36) |
> meta-DKF          | 2.34(1.57)e-2 | 0.47(0.41) | 3.77(3.84) | 0.88(0.22) |
>
> We are very thankful to the reviewer to push us to devise this SNP-based meta-formulation for the LVMs in Fig. 1A. This adds a strong baseline to our work, and also further supports our analyses of the fundamental pros and cons of the LVMs formulations in Fig. 1A and B. They are added to the revised manuscript.
>
> **Variational approximation of c given by the query vs. support sets**
>
> If the reviewer is asking about the variational approximation of $c$ in the last KL term of equation (8): yes, at meta-training time, the query example is simply combined with the support set, and their embedded features are averaged. This is done for each individual query example at a time, and there is no weighting between the support set vs. the query example.
>
> The above only occurs at meta-training time. At test time, the set-embedding of $c$ comes only from the support set.

---

> > ### Author Response · Authors · 2022-11-29
> > **Reminder to Reviewer EkYh**
> >
> > Dear Reviewer EkYh
> >
> > Thank you again for your constructive feedback. We hope that you have had time to go through our response and revised manuscript in addressing your previous comments. We would like to follow up to see if you have additional comments for further discussion.
> >
> > Best regards,
> >
> > Authors

---

### Official Review · Reviewer_Chgq · 2022-11-03

**Confidence:** 3
**Correctness:** 4
**Technical Novelty And Significance:** 3
**Empirical Novelty And Significance:** 2
**Recommendation:** 8

**Clarity, Quality, Novelty And Reproducibility:**

This work seems original, though there are similar efforts in related areas, this method both unifies and justifies the approach with background/prior work. The addition of few shot setups in dynamics learning is of broad interest, and well supported by the proposed model.

"We then present the first framework of few-shot forecasting for high-dimensional time-series" in the abstract is a bit of an overclaim (given extensive prior work on few, one, and zero shot learning for video and audio generation, as well as the broader relationship of prompting and transformers to few shot settings), and in this case not necessary for the core method to be novel and of interest.

The appendix is extensive, and related links are helpful. Code is included, and the effort in terms of reproducibility of the approach is top notch. In addition the authors detail what packages were used for the benchmark comparisons, which is critical to any followup comparisons.

**Strength And Weaknesses:**

Strengths:

Strong presentation of the method, background, and related work. Logical extension to meta/few-shot setting, with direct references to how and why the architecture is logical. Results compare favorably with benchmarks, benchmarks are pretty thoroughly set up and tested on the chosen datasets.

Weaknesses:

Figure 4 color scheme is pretty difficult to discern, specifically two slightly different shades of green. Generally Figures 2, 3, and 4 (and to some extent 5) have many graphical results which should probably be condensed to a table, with a small example of frames from 1 or 2 chosen methods. More extensive videos/gifs can be linked in a project page, or in drive links like those provided in the appendix. Generally it would be nice to see improvements to the experimental results presentation in this section.

The comparisons are thorough but FiVO seems to be missing - I do not think a full comparison is totally necessary to show the value here, but some discussion or citation is probably in order given the close relation of FiVO to the related benchmark models tested https://arxiv.org/abs/1705.09279 .

Addition of another experiment beyond the existing would broaden the pool of potential readers. Particularly there are a variety of benchmarks in the related work (midi music, speech, handwriting) that could make compelling comparison points. Despite the discussion of multi-dimensional data, it is worth considering or testing low-dimensional forecasting, as the formulation of few shot dynamics has direct relevance in that subfield, and there are many benchmarks in GluonTS which DeepAR used. Multi-speaker TTS (or audio modeling ala RAVE https://arxiv.org/abs/2111.05011 ) is another compelling area, but may be out of scope due to the limits of the reviewer/author feedback window. Online handwriting is at least multi-dimensional, and has many prior applications related to stylistic generation (see VRNN or https://arxiv.org/abs/2110.02891 ). The experiments in the paper show the value of the method, but the scope of experiments seems narrow compared to some of the prior work, making it difficult to asses the full impact in the design choices of this method.

The cardiac electrical dynamics seems to be an interesting task, but hard to understand or compare due to limited space. Consider redistributing importance (in terms of writing space allocated) between these experiments. However, a problem may arise given the comparison method - what do *failures* look like for this model in the cardiac electrical case? Are they disastrous, or tolerable for the setting? Given the strong baseline approach (PVH), the errors from this method are probably qualitatively different - discussion of what types of failures could occur seem important for practical applications.

It would be nice to mention use of exemplar based methods in static VAE settings, such as Exemplar VAE https://proceedings.neurips.cc/paper/2020/hash/63c17d596f401acb520efe4a2a7a01ee-Abstract.html , probably as related work.

The primary points I would like to see addressed are:
1) modified presentation of results in figures 2/3/4/5, with a focus on moving numerical results into tables instead of bar charts for easier comparison - as it stands these results are difficult to parse.
2) Further work on the experiments - either introducing an additional task with more common ground with the related work, or a deeper dive into the results on cardiac electrical dynamics, what the results mean, and what the potential impact of those results could be on the relevant application field.
Addressing these points (either, or both - in whole or in part) would improve the quality of the presentation, and justify a higher score.

As mentioned in the paper, comparison to Lagrangian and Hamiltonian networks could be of interest if the focus remains on physics-based experimental benchmarks.

Some minor typos to fix:
"abstracts a a latent", paragraph 2
"Deepar" -> "DeepAR"

**Summary Of The Paper:**

Sequential Latent Variable Models for Few-Shot High-Dimensional Time-Series Forecasting introduces a sequential latent variable model for meta-learning various high-dimensional time-series. This model allows for learning diverse dynamics, and adapting in a few-shot setting during evaluation, evaluating on a variety of benchmark datasets against known models from prior literature.

**Summary Of The Review:**

This paper introduces an interesting method, and both the background writing and method discussion make clear the relevant materials, and why this method is interesting. However the scope of the experiments, along with the presentation of the experiments, currently limit the potential interest of this work.

---

> ### Author Response · Authors · 2022-11-19
> **Response to Reviewer Chgq**
>
> We thank the reviewer for your appreciation and encouraging feedback on our work.
>
> **Presentation of results**
>
> Thanks for the suggestion. We have re-organized all of our experimental results (including newly added ones in this rebuttal) into tables (Tables 1-3 and figure 2), with visuals of only a small number of selected examples representing the key LVM formulations in Fig 1 and their meta-extensions (moving the rest to Appendix). We also changed the color code for Fig 4A (now Fig 3A) for better readability.
>
> **Additional experiments**
>
> We have responded to both aspects of the reviewer’s suggestions.
>
> We explored a variety of relevant datasets to identify which we may be able to customize our model architecture to within the rebuttal period. We eventually selected a dataset of synthetic turbulent flow dynamics with variance in dynamics arising from the buoyant force acting on the fluid. This dataset was used in a recent NeurIPS 2022 paper. Please see bullet 4 of our overall response for more details.
>
> In the meantime, we have revised Section 6 to provide a deeper dive into the results from cardiac electrical dynamics about what is being modeled, what failure looks like, and what are the potential impacts on the application field (especially in comparison to the PVH).
>
> We also would like to point the reviewer to a significant addition of the meta-extensions of the main sequential LVM baselines, which is summarized in bullets 1-3 of our overall response.
>
> **Discussion and relation to FiVO and Exemplar VAE**
>
> We thank the reviewer for bringing up these two works. We have added a relevant discussion to the original manuscript.
>
> FiVO is highly related to the class of sequential LVMs modeling the poster distribution of latent states $z_k$ over time, as summarized in Fig 1A of our paper. Similar to how particle filters extend Kalman filters beyond linear transition functions and Gaussian density assumption, we feel that FiVO can be seen as a more sophisticated extension of DKF. We added citation and discussion of FiVO to Section 2 when discussing Fig 1A formulation of the sequential LVMs.
>
> Exemplar VAE (and exemplar-based generative models) has an interesting high-level relation with the presented (or general feed-forward) meta-model, if we view the few-shot context examples as the exemplar (from which a latent code is obtained and then transformed into a query example). We added a discussion of this relation in Related Works when discussing general few-shot learning.

---

> > ### Author Response · Authors · 2022-11-29
> > **Reminder to Reviewer Chgq**
> >
> > Dear Reviewer Chgq
> >
> > Thank you again for your constructive feedback. We hope that you have had time to go through our response and revised manuscript in addressing your previous comments. We would like to follow up to see if you have additional comments for further discussion.
> >
> > Best regards,
> >
> > Authors

---

> > > ### Comment · Reviewer_Chgq · 2022-12-12
> > > **Reply to Authors post Rebuttal**
> > >
> > > After seeing the updated responses to reviewers and the intensive revisions to the paper, I have updated my score. These changes, accomplished in the limited time of the rebuttal phase, strengthen the presentation of the work and should be beneficial to interested readers and followup researchers.

---

### Author Response · Authors · 2022-11-19
**Summary of major additions**

We thank the reviewers for their constructive feedback. While detailed responses are provided to each reviewer, here we highlight several major additions that add further evidence about the contribution of our work.

**Meta-extensions of representative sequential LVM baselines**: in this paper, we unify existing sequential LVMs to two general formulations – those based on the modeling of system state $q(z_k)$ at each time $k$ (Fig. 1A), and those based on modeling of system parameters q(c) that underlying the entire dynamics (Fig. 1B). Although meta-models for these sequential LVMs do not exist in literature, in this rebuttal we have added the meta-extensions for models representing each formulation: DKF for the former, and the latent GRU-res for the latter.

**Feed-forward meta-models**: Our presented framework (Fig 1C) is based on a feed-forward meta-model for the latter LVM formulation (represented by the latent GRU-res). Thus, we added comparisons to a similar feed-forward meta-model extension to the former LVM formulation (represented by the DKF). This also represents an extension of the Sequential Neural Process to the forecasting problem under consideration (as suggested by reviewer EkYh). Experiment results are added throughout Table 1-2 & Fig 2 of the revised manuscript (see rows of “meta-DKF”), providing further evidence that this type of sequential LVMs – while good at modeling & reconstructing time-sequences – are limited in forecasting without near-time observations. The use of meta-learning does not overcome this limitation.

**Optimization-based  (MAML-based) meta-models**: As suggested by reviewer wdac, we also experimented with extending MAML to DKF and latent GRU-res. While the implementation was relatively straightforward, both models faced issues of training instability and lack of convergence despite our efforts in incorporating various suggestions given in literature to address the optimization issues and testing various hyper-parameters  [2]. We believe that this is consistent with meta-learning literature [1] that has reported issues of MAML when scaling up to large input dimensionality and more complex modeling architectures [1], attributed to factors such as vanishing gradient issues over the complex computation graph and the difficulty of tuning the inner-loop optimization rate [2].  The lack of literature reporting MAML extension to VAE-based sequential LVMs may further attest to these challenges.  With this, we suspect that the extension of MAML for high-dimensional sequential LVMs is a non-trivial pursuit – we hope this provides further evidence that the presented meta-learning framework for few-shot sequential LVMs is by no means incremental.

**Addition of new benchmark experiments on complex physics**: In response to reviewer Chgq and wdac, we have added experiments on an additional dataset of synthetic turbulent flow dynamics with variance in the buoyant force acting on the fluid. This dataset was used in a recent NeurIPS 2022 paper [3] and contains 25 buoyancy forces considered to be the varying learning tasks. Similar to our presented work, this recently published work reports an autoregressive forecasting framework uses a “meta-model” to learn an embedding to condition the forecasting model. This “meta-model”, however, is trained independently before training the forecasting model to infer tasks (representing different dynamics), and it leverages a weak-supervision loss that utilizes known or able-to-be-estimated task parameters. Given a time-series to forecast, this trained task-encoder then uses the observed time frames of the series to infer the task (whereas our feed-forward meta-model obtains embedding from support sets distinct from the query series, and the meta-model is trained end-to-end, as in typical meta-learning). Therefore, despite the shared idea of conditioning the forecaster with dynamic-specific embedding, these two approaches are substantially different (let alone the base LVM vs. autoregressive formulation).

Following the experimental setup in [3], we demonstrated that our presented meta-GRU was able to outperform the results reported in [3] despite using a smaller number of observed frames to forecast the same roll-out window: the presented meta-GRU model obtained a rooted MSE (RMSE) of $0.26\pm0.05$ on seen dynamics versus $0.42\pm0.01$ in [3], and $0.49\pm0.05$ on unseen dynamics versus  $0.51\pm0.02$ reported in [3]. This addition of experiments thus not only adds another benchmark with more complex dynamics, but also adds comparison to a most recent baseline to the presented model. Results are added to Section 6 of the revised manuscript.

[1] A User’s Guide to Calibrating Robotic Simulators (https://proceedings.mlr.press/v155/mehta21a.html)

[2] How to train your MAML (https://arxiv.org/abs/1810.09502)

[3] Meta-learning dynamics forecasting using task inference (https://arxiv.org/abs/2102.10271)

---

### Author Response · Authors · 2022-12-05
**Thanks again for your time and efforts**

Dear reviewers,

Thanks again for your time and efforts in reviewing our paper. As the deadline for discussion is approaching, we hope that you had time to revise our revised manuscript and responses. We look forward to understanding if our response/revision has properly addressed your concerns, and if you have further concerns that we could address before the discussion period ends.

Best regards,
Authors

---

### Decision · Program_Chairs · 2023-01-20

**Decision:**

Accept: notable-top-25%

**Justification For Why Not Higher Score:**

The limited technical novelty likely precludes it from receiving an oral relative to other papers.

**Justification For Why Not Lower Score:**

Particularly after the rebuttal, the experiments section is now quite strong, and is of interest to the community.

**Metareview: Summary, Strengths And Weaknesses:**

The authors propose Sequential Latent Variable Models (SLVM) for Few-Shot High-Dimensional Time-Series Forecasting, a method to learning diverse dynamics with meta-learning. The ideas will be of interest to the community, the presentation is strong, and the experiments thorough. The only minor issue is that the fundamental idea is only relatively novel, but on the balance this is a good paper.

**Note From Pc:**

if the above contains the word "oral" or "spotlight" please see: "oral" presentation means -> notable-top-5% and "spotlight" means -> notable-top-25%. As stated in our emails, we are disassociating presentation type from AC recommendations